# USP13 antagonizes gp78 to maintain functionality of a chaperone in ER-associated degradation

Yanfen Liu[1], Nia Soetandyo[1], Jin-gu Lee[1], Liping Liu[1], Yue Xu[1], William M Clemons Jr[2], Yihong Ye[1]*

[1]Laboratory of Molecular Biology, National Institute of Diabetes and Digestive and Kidney Diseases, National Institutes of Health, Bethesda, United States; [2]Division of Chemistry and Chemical Engineering, California Institute of Technology, Pasadena, United States

**Abstract** Physiological adaptation to proteotoxic stress in the endoplasmic reticulum (ER) requires retrotranslocation of misfolded proteins into the cytoplasm for ubiquitination and elimination by ER-associated degradation (ERAD). A surprising paradox emerging from recent studies is that ubiquitin ligases (E3s) and deubiquitinases (DUBs), enzymes with opposing activities, can both promote ERAD. Here we demonstrate that the ERAD E3 gp78 can ubiquitinate not only ERAD substrates, but also the machinery protein Ubl4A, a key component of the Bag6 chaperone complex. Remarkably, instead of targeting Ubl4A for degradation, polyubiquitination is associated with irreversible proteolytic processing and inactivation of Bag6. Importantly, we identify USP13 as a gp78-associated DUB that eliminates ubiquitin conjugates from Ubl4A to maintain the functionality of Bag6. Our study reveals an unexpected paradigm in which a DUB prevents undesired ubiquitination to sharpen substrate specificity for an associated ubiquitin ligase partner and to promote ER quality control.

*For correspondence: yihongy@mail.nih.gov

**Competing interests:** The authors declare that no competing interests exist.

**Reviewing editor**: Randy Schekman, Howard Hughes Medical Institute, University of California, Berkeley, United States

## Introduction

In eukaryotic cells, secretory and membrane proteins are processed in the endoplasmic reticulum (ER). In this protein maturation process, nascent polypeptides need to pass stringent checkpoints monitored by a sophisticated quality control program. Only correctly folded proteins can exit the ER and reach their final destinations, whereas terminally misfolded or unassembled polypeptides are retained and eliminated by the ER-associated degradation (ERAD) system (*Vembar and Brodsky, 2008*; *Smith et al., 2011*). This mechanism not only adapts cells to proteotoxic stress (*Balch et al., 2008*), but also fine-tunes signaling processes essential for the development of multicellular organisms (*Ron and Walter, 2007*).

ERAD requires retrotranslocation of misfolded polypeptides across the membrane for ubiquitination by one of a few ER-associated ubiquitin conjugating systems, each comprising a ubiquitin ligase (E3) and a cognate conjugating enzyme (E2) (*Hirsch et al., 2009*; *Ye and Rape, 2009*). Ubiquitin ligases are a key component of the cellular ubiquitination machinery that acts at the final step of ubiquitin transfer to ensure only certain substrates are selectively ubiquitinated in both temporally and spatially regulated manners (*Deshaies and Joazeiro, 2009*). Substrate specificities of E3s are thought to be an intrinsic property of these enzymes, but whether it can be modulated by additional factors is an open question. In ERAD, substrate ubiquitination occurs on the cytosolic side of the ER membranes, immediately after substrates have emerged from the ER lumen. This modification serves as a signal to recruit the p97/VCP ATPase, leading to the extraction of substrates from the ER membrane

**eLife digest** Cells make proteins inside a structure called the endoplasmic reticulum. However, some of these proteins cannot fold into the correct shape, so cells rely on a process called the ERAD pathway to degrade and eliminate these faulty proteins. First, however, the misfolded proteins must be moved from the endoplasmic reticulum to the main body of the cell (the cytosol).

The process by which the misfolded proteins are moved through the membrane that encloses the endoplasmic reticulum is complex, with 'ERAD machinery proteins' playing an important role. Among them, a series of enzymes called E3 ligases 'tag' the faulty proteins with a small protein called ubiquitin, and a complex called the proteasome then recognizes and degrades those proteins that have been tagged with ubiquitin. However, it is not clear why the E3 ligases that tag the misfolded proteins with ubiquitin don't also tag the machinery proteins that from complexes with the faulty proteins.

Now Liu et al. have used a combination of biochemical and genetic tools to shed light on this puzzle by studying the interaction of gp78—which is an E3 ligase—and USP13, an enzyme that opposes the actions of the E3 ligases by removing ubiquitin. Liu et al. showed that gp78 can indeed tag certain machinery proteins with ubiquitin, which would stop the removal of misfolded proteins from the endoplasmic reticulum. However, USP13 opposed the action of gp78, thus allowing the removal to continue.

It has been known for some time that enzymes with opposing roles—the addition and removal of ubiquitin—can work together, but the biological significance of this phenomenon was not fully understood. The work of Liu et al. suggests that USP13 makes the elimination of misfolded proteins more efficient by ensuring that gp78 only tags those proteins that are misfolded: it does this by removing ubiquitin from proteins that should not have been tagged. A similar phenomenon is known to occur in genetics during DNA replication, with the enzyme complex that replicates the DNA including an enzyme that performs a proofreading role.

(*Bays et al., 2001*; *Ye et al., 2001*; *Jarosch et al., 2002*; *Flierman et al., 2003*). Additionally, it facilitates subsequent targeting of retrotranslocated substrates to the proteasome (*Vembar and Brodsky, 2008*).

In mammalian cells, the multi-spanning membrane protein gp78 is one of the most extensively studied E3s in ERAD (*Fang et al., 2001*; *Song et al., 2005*; *Jo et al., 2011*; *Chen et al., 2012*). gp78 is homologous to Hrd1p, an ERAD-specific E3 in yeast proposed to form a 'retrotranslocon' that channels misfolded proteins out of the ER (*Carvalho et al., 2010*). Similar to Hrd1p, gp78 also forms a large membrane complex that communicates with machinery proteins on both sides of the membrane (*Zhong et al., 2004*; *Christianson et al., 2011*), establishing it as a master regulator of retrotranslocation. On the luminal side, ER chaperones and lectins recognize and deliver misfolded proteins to a gp78-containing complex for retrotranslocation. In the cytosol, gp78 uses a VIM (VCP-interacting motif) segment to bind p97/VCP (*Ballar et al., 2006*). In addition, it has a CUE (coupling ubiquitin to ER degradation) domain that recruits a multiprotein complex comprising Bag6 and its cofactors Ubl4A and Trc35 (*Chen et al., 2006*; *Wang et al., 2011*).

Bag6 contains an unusual chaperone 'holdase' activity, capable of maintaining retrotranslocated polypeptides in a soluble state to enhance their turnover (*Wang et al., 2011*). Moreover, Bag6 was reported to interact transiently with the proteasome (*Minami et al., 2010*), raising the possibility that it may facilitate substrate hand over from the gp78–p97 complex to the proteasome. Importantly, this 'holdase' activity can also act in conjunction with other ubiquitin ligases to help degrade defective cytosolic and mislocalized proteins (*Minami et al., 2010*; *Hessa et al., 2011*), establishing Bag6 as a major E3-associated chaperone in protein quality control (*Lee and Ye, 2013*).

In addition to ubiquitin ligases and associated factors, recent studies have implicated deubiquitinases (DUBs), enzymes that cleave ubiquitin conjugates, as important regulators of ERAD. However, unlike ubiquitin ligases, the functions of DUBs in ERAD are obscure. Many DUBs involved in ERAD associate with p97/VCP either directly or indirectly. These include YOD1 (*Ernst et al., 2009*), ataxin-3 (*Wang et al., 2006*; *Zhong and Pittman, 2006*), USP25 (*Blount et al., 2012*), USP13, USP50, and VCPIP1 (*Sowa et al., 2009*). Interestingly, although deubiquitination has often been reported to cause removal of the

proteasome targeting signals and therefore inhibition of protein turnover, many p97-associated DUBs apparently serve as positive regulators in ERAD (*Wang et al., 2006*; *Ernst et al., 2009*; *Sowa et al., 2009*). Several models have been proposed to explain these surprising findings, but experimental data in support of any of these models are lacking (*Liu and Ye, 2012*).

In this study, we characterize a p97-associated deubiquitinase termed USP13. Our study reveals an unexpected interaction between USP13 and the ERAD-specific ubiquitin ligase gp78. Despite having opposing activities, USP13 can cooperate with gp78 to promote ERAD. Mechanistically, USP13 plays a crucial role in fine-tuning the substrate specificity of gp78 as in its absence ubiquitin chains assembled by gp78 can accumulate on an ERAD machinery component—the Bag6-Ubl4A-Trc35 complex that is associated with proteolytic processing of Bag6 and altered interaction between Bag6 and SGTA. Together, these results established that the activity of an E3 ubiquitin ligase can be safeguarded by an accompanying deubiquitinase to enhance its specificity.

## Results

### USP13 and USP5 differ in both activities and interactors

USP13 shares ~80% sequence similarity to USP5 (*Figure 1—figure supplement 1A*), an isopeptidase that preferentially cleaves unanchored ubiquitin chains (*Reyes-Turcu et al., 2006*). However, only USP13, but not USP5 has been implicated in ERAD. To understand how these homologous enzymes can have distinct functions, we first purified recombinant USP13 and USP5 from both mammalian cells and *Escherichia coli*. We tested their ability to bind free ubiquitin and to cleave ubiquitin-AFC (Ub-AFC) in vitro. Interestingly, USP5 readily bound to free ubiquitin and cleaved Ub-AFC with high efficiency, whereas USP13 was almost completely inactive (*Figure 1A*, *Figure 1—figure supplement 1B,C*). USP13 could not efficiently cleave K48- or K63-linked di-ubiquitin either (data not shown). We next generated and purified USP13 mutants in each of which a fragment in USP13 was replaced by a corresponding segment in USP5 (*Figure 1B*). Ub-AFC assay showed that the chimera (USP13/5-4) in which the first 722 amino acids of USP13 were replaced with a corresponding segment in USP5 still had no deubiquitinating activity. However, further incorporation of a USP5 segment encompassing the second ubiquitin-associated (UBA) domain (amino acids 717–781) to USP13 (USP13/5-5) led to an activity similar to that of USP5 (*Figure 1B,C*). These data suggest that USP13 and USP5 are distinct deubiquitinases and the second UBA domain accounts for the differential activities of these enzymes.

USP13, but not USP5 was reported to bind p97 (*Sowa et al., 2009*), suggesting that these enzymes also differ in binding partners. To understand the role of USP13 in ERAD, we characterized its interaction with p97 using an in vitro binding assay. We immobilized purified GST-USP13 or GST-USP5 on glutathione beads and incubated the beads with recombinant p97. As a positive control, we used GST-VIMP, a known p97 interactor (*Ye et al., 2004*). As expected, the glutathione beads containing GST-VIMP pulled down p97 efficiently. By contrast, neither GST-USP13 nor GST-USP5 bound p97 (*Figure 1D*). However, when glutathione beads containing the same set of proteins were incubated with whole cell extracts, both VIMP and USP13 co-precipitated with endogenous p97, but USP5 still failed to bind p97 (*Figure 1E*). These results indicate that USP13, but not USP5, interacts with p97 through an adaptor(s) in cells, which explains why USP13, but not USP5 can function in ERAD.

### USP13 interacts directly with gp78 in ERAD

We used the GST pull-down assay to screen a collection of known p97 interactors in order to identify factor(s) that promotes the USP13-p97 interaction. We expressed these proteins in cells individually and examined the binding of endogenous p97 to GST-USP13. Compared to the control (cells transfected with an empty vector), expression of several UBX domain-containing p97 cofactors including ASPL, SAK1, and UbxD1 reduced the p97-USP13 interaction (*Figure 2—figure supplement 1A*, lanes 3, 5, 7 vs lane 1), suggesting that these factors may compete with USP13 for p97 binding. By contrast, several other factors enhanced the p97-USP13 interaction. Among them, the most significant enhancer was gp78 (lane 16), a ubiquitin ligase known to mediate retrotranslocation and ubiquitination of many ERAD substrates (*Fang et al., 2001*; *Song et al., 2005*; *Christianson et al., 2011*; *Jo et al., 2011*; *Chen et al., 2012*).

Several lines of evidence confirmed that gp78 could bind to USP13 directly and promote its interaction with p97. Firstly, GST-USP13 co-precipitated with both endogenous and overexpressed gp78 in addition to p97 (*Figure 2A*, lanes 4, 7). By contrast, USP5 did not bind endogenous gp78, and only a

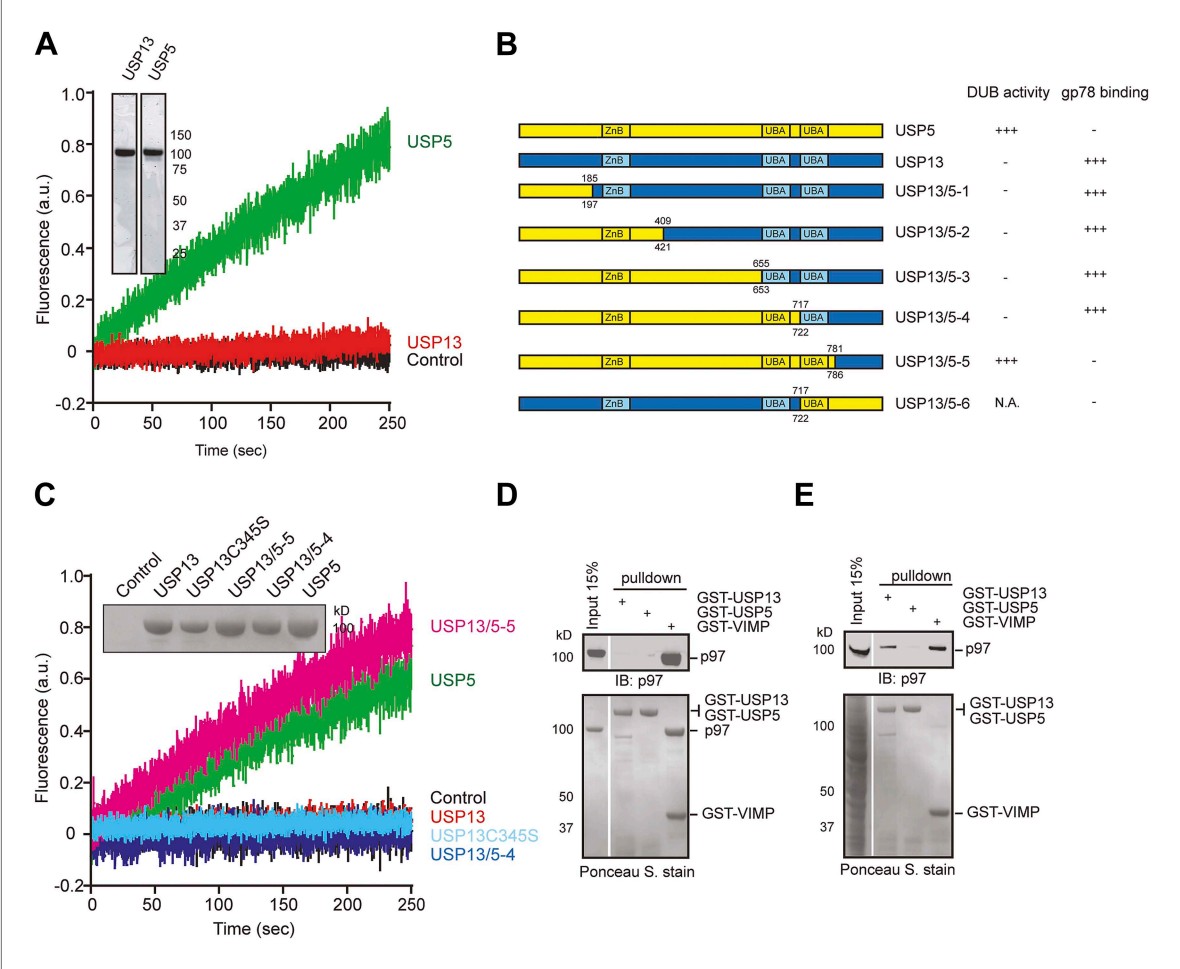

**Figure 1**. USP13 and USP5 have distinct activities despite sequence homology. (**A**) Purified USP13 is inactive whereas USP5 is active. The activities of the purified USP5 and USP13 were measured by the Ub-AFC assay. The Coomassie blue-stained gels show the purified proteins. (**B**) A schematic illustration of the USP13-USP5 chimeras and a summary of their deubiquitinating activities. (**C**) The deubiquitinating activity of the indicated USP13-USP5 chimeras as measured by the Ub-AFC assay. The USP13 catalytic inactive mutant C345S was used as a negative control. The inset shows the purified proteins. (**D**) USP13 does not bind p97 directly. The indicated GST-tagged proteins were immobilized and incubated with recombinant p97. The precipitated proteins were analyzed by immunoblotting (IB). (**E**) USP13, but not USP5, binds p97 through an adaptor. As in **D**, except that a whole cell extract was used in replace of p97.

The following figure supplements are available for figure 1:

**Figure supplement 1**. USP13 and USP5 have distinct deubiquitinating activities.

negligible amount of overexpressed gp78 could be pulled down by GST-USP5 (lanes 5, 8). Secondly, depletion of gp78 by shRNA-mediated gene knockdown or gene deletion reduced the USP13-p97 interaction by threefold to fourfold (*Figure 2B*, *Figure 2—figure supplement 1B*). An endogenous interaction between USP13 and gp78 was also detected (*Figure 2C*). Moreover, GST-tagged gp78 cytosolic domain (gp78c) could bind recombinant USP13 in vitro (*Figure 2D*). These results demonstrated a specific and direct interaction between USP13 and gp78 that facilitates USP13-p97 association. The data also suggest that the interaction of USP13 with p97 is mediated by several cellular factors with gp78 being a major one.

We next tested the interaction of gp78 with the various USP13/5 chimeras to map the domain in USP13 responsible for gp78 interaction. Co-immunoprecipitation showed that the chimeras defective in cleaving Ub-AFC also behaved similarly to USP13 in binding gp78 (*Figure 2E*, lanes 3–7). By contrast, USP13/5-5 had similar DUB activity as USP5, and both the proteins only co-precipitated with marginal amount of gp78 (lanes 2, 8). It appears that the second UBA domain is crucial for USP13 to bind gp78. To

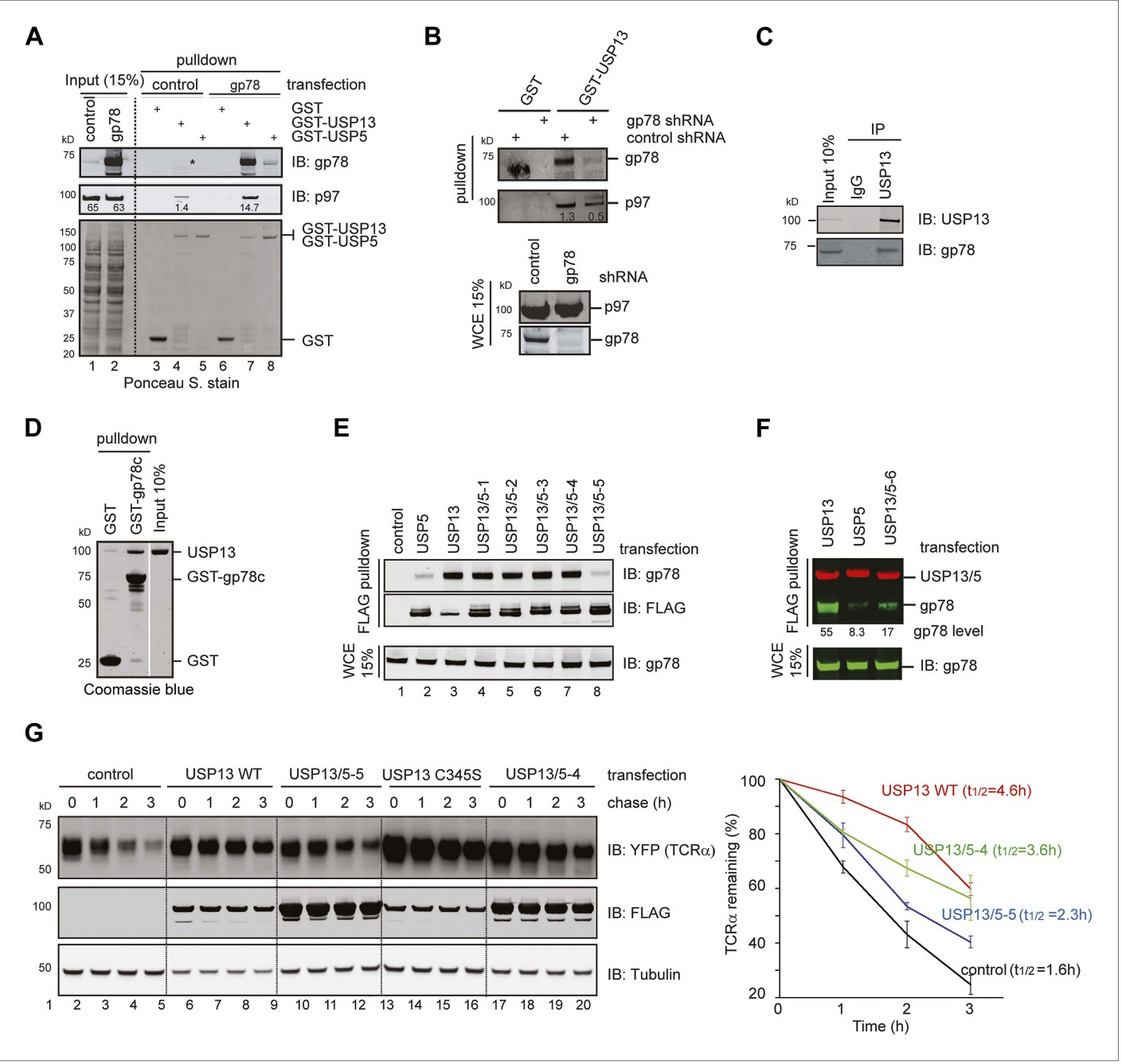

**Figure 2**. USP13 forms a complex with the ERAD E3 gp78. (**A**) gp78 enhances the interaction of USP13 with p97. Cell extracts prepared from control or gp78-expressing cells were incubated with glutathione beads containing the indicated proteins. The asterisk indicates a fraction of endogenous gp78 that is co-precipitated with GST-USP13. The numbers indicate the levels of p97. (**B**) Depletion of gp78 reduces the interaction of USP13 with p97. Whole cell extracts (WCE) from control or gp78 knockdown cells were incubated with glutathione beads containing either GST or GST-USP13. (**C**) An interaction between endogenous USP13 and gp78. Whole cell extract was subject to immunoprecipitation (IP) with either control or anti-USP13 antibodies. (**D**) USP13 binds the gp78 cytosolic domain (gp78c) directly. The indicated GST-tagged proteins were immobilized and incubated with recombinant USP13. (**E** and **F**) Interactions of the indicated USP13-USP5 chimeras with gp78. WCEs from cells expressing the indicated DUB chimeras and gp78 were subject to IP with FLAG beads. (**G**) ERAD inhibition by overexpressed USP13 is dependent on gp78 interaction. TCRα-YFP cells transfected with the indicated USP13-expressing plasmids were treated with cycloheximide for the indicated time points. Cell extracts were analyzed by immunoblotting. The graph represents quantification of three independent experiments. Error bars, SD (n = 3).

The following figure supplements are available for figure 2:

**Figure supplement 1**. Identification of factors that influence the USP13-p97 interaction.

further test this idea, we created an additional construct (USP13/5-6) that had the amino acid sequence downstream of the first UBA domain of USP13 replaced with the corresponding segment from USP5 (*Figure 1B*). Co-immunoprecipitation showed that like USP5, this mutant also bound gp78 with significantly reduced affinity when compared to USP13 (*Figure 2F*). Collectively, these results suggest that the second UBA in USP13 inhibits its DUB activity, but at the same time allows it to gain interaction with gp78.

When the degradation of the model ERAD substrate TCRα was examined in cells overexpressing either USP13 or the USP13 variants, we found that wild-type USP13 significantly increased the half-life of TCRα (from 1.6 hr to 4.6 hr). Because this phenotype was similarly observed in cells expressing the catalytically inactive USP13 mutant (USP13 C345S) (*Figure 2G*), as well as in USP13 knockdown cells (see below, *Figure 3C*), we concluded that overexpressed USP13 had a dominant negative effect on ERAD. This was not entirely surprising as several reports had shown similar dominant negative activities with overexpression of other wild-type ERAD factors, presumably because the overexpressed proteins disrupt the normal stoichiometry of endogenous complexes. Consistent with this interpretation, we found that the ERAD inhibitory activity of USP13 was strongly dependent on gp78 binding because USP13/5-4, a chimera bound gp78 similarly to wild-type USP13, inhibited TCRα degradation ($t_{1/2}$ = 3.6 hr), whereas the chimera (USP13/5-5) that had reduced affinity to gp78 only slightly increased the half-life of TCRα ($t_{1/2}$ = 2.3 hr). The requirement of gp78 interaction for ERAD inhibition by ectopic USP13 indicates that the interaction of USP13 with gp78 is functionally relevant to ERAD.

## USP13 regulates ERAD by promoting substrate solubility

To understand how USP13 regulates ERAD, we characterized the ERAD defects in USP13 knockdown cells. First, we used two USP13 specific shRNAs to knock down USP13 in a cell line stably expressing TCRα-YFP. USP13 depletion significantly increased the levels of TCRα-YFP, as revealed by both immunoblotting and flow cytometry analyses (*Figure 3A,B*). This phenotype could be attributed to increased TCRα stability as demonstrated by cycloheximide (CHX) chase experiments (*Figure 3C*, lanes 5–8 vs lanes 1–4). Because depletion of DUBs may cause ubiquitin deficiency that can indirectly inhibit proteasomal degradation (*Reyes-Turcu et al., 2009*), we generated a cell line expressing a TCRα variant lacking the signal peptide (ΔsTCRα-YFP). This TCRα mutant is not targeted to the ER. Consequently, it is degraded by an ERAD-independent mechanism (*Fiebiger et al., 2004*). Notably, USP13 knockdown had no effect on the steady state level of ΔsTCRα-YFP (*Figure 3D*). We conclude that USP13 is specifically required for efficient turnover of ERAD substrates.

We next determined which step in ERAD is blocked by USP13 knockdown. To this end, we analyzed the glycosylation status of TCRα in USP13-depleted cells that had been exposed to a proteasome inhibitor. TCRα is a glycosylated type I membrane ERAD substrate that undergoes deglycosylation upon retrotranslocation (*Yu et al., 1997*). Accordingly, in cells treated with the proteasome inhibitor MG132, TCRα is accumulated in two forms, an ER-localized, glycosylated precursor and a deglycosylated intermediate that has emerged into the cytosol. The level of deglycosylated TCRα therefore can be used to gauge the retrotranslocation activity. As anticipated, knockdown of p97, the retrotranslocation-driving ATPase, reduced deglycosylated TCRα in MG132-treated cells due to inhibition of retrotranslocation (*Figure 3E*, lane 3 vs 1). By contrast, USP13 depletion did not affect deglycosylation of TCRα (lane 2 vs 1), suggesting that USP13 promotes ERAD downstream of p97.

Because we recently established the Bag6-Ubl4A-Trc35 complex as a key mediator that channels substrates from the sites of retrotranslocation to the proteasome, we asked whether USP13 may cooperate with Bag6 to promote substrate delivery to the proteasome. As demonstrated previously (*Wang et al., 2011*), depletion of Bag6 caused accumulation of TCRα-YFP in aggregates in a fraction of cells (~30%) due to lack of chaperoning during the proteasome targeting steps. TCRα in these cells was resistant to extraction by the non-ionic detergent NP40 (*Figure 3F,G*). Intriguingly, we observed accumulation of TCRα in aggresome-like punctae in similar number of the TCRα-YFP-expressing cells under USP13 knockdown conditions (*Figure 3F*), whereas in control cells, TCRα-containing aggregates was detected only in ~8% of the cells. Immunoblotting confirmed that USP13 depletion caused more TCRα to partition into the NP40-insoluble fractions (*Figure 3G*) compared to control cells. For comparison, knockdown of p97 resulted in greater levels of TCRα stabilization, but only a small fraction of TCRα was present in the NP40-insoluble fraction (*Figure 3—figure supplement 1A*, lane 8 vs 4). Because p97 depletion caused TCRα to accumulate primarily inside the ER, our data suggest that TCRα forms aggregates in USP13-depleted cells most likely after it has entered the cytosol.

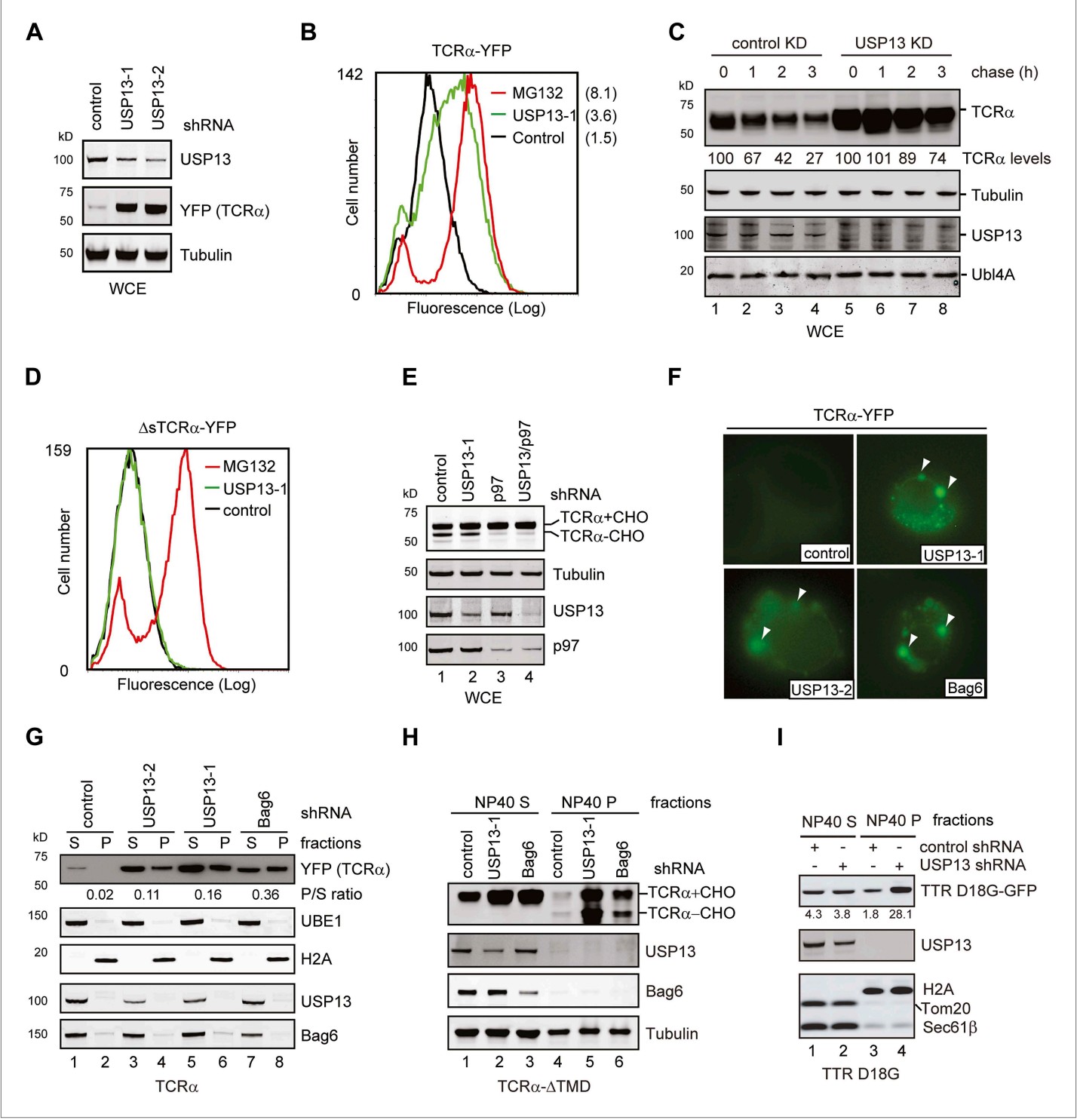

**Figure 3**. USP13 is required to maintain the solubility of a retrotranslocation substrate. (**A–C**) USP13 knockdown stabilizes the ERAD substrate TCRα. (**A**) HEK293 cells stably expressing TCRα-YFP were transfected with the indicated shRNA-expressing plasmids. Proteins in the whole cell extracts (WCE) were analyzed by immunoblotting. (**B**) Control or USP13 knockdown cells expressing TCRα-YFP were analyzed by flow cytometry. The proteasome inhibitor MG132 was used as a positive control. The numbers indicate the calculated X-mean values. (**C**) The stability of TCRα-YFP in control and USP13 knockdown cells was analyzed by CHX chase. The number indicated the relative amount of TCRα-YFP averaged from two independent experiments. (**D**) USP13 knockdown does not stabilize a non-ERAD substrate. As in **B**, except that cells stably expressing ΔsTCRα-YFP was used. (**E**) USP13 functions at a step downstream of p97-mediated dislocation. The TCRα-YFP-expressing cells treated with the indicated shRNA-expressing plasmids were exposed to
*Figure 3. Continued on next page*

*Figure 3. Continued*

the proteasome inhibitor MG132 (10 μM, 15 hr). TCRα+CHO, glycosylated TCRα; TCRα-CHO, deglycosylated TCRα. (**F**) TCRα-YFP-containing aggregates accumulate in USP13- and Bag6-depleted HEK293 cells. Shown are representative cells transfected with a TCRα-YFP-expressing plasmid together with the indicated shRNA constructs. Arrowheads indicate TCRα-YFP punctae formed upon depletion of USP13 or Bag6. (**G**) TCRα-YFP-containing aggregates in USP13-depleted cells are resistant to extraction by NP40. As in **F**, except that cells were subject to sequential extraction, first by an NP40-containing lysis buffer, then by the Laemmli buffer. UBE1 and H2A serve as the markers for the NP40 soluble (S) and insoluble (P) fractions, respectively. Note that the NP40-insoluble fractions were loaded two times of the soluble fractions. The numbers indicate the ratio of TCRα in the pellet vs the soluble fraction. (**H** and **I**) As in **G**, except that cells expressing the indicated ERAD substrates were used. The numbers in **I** indicate the levels of TTR D18G. Note that a Bag6 antibody against the C-terminus of Bag6 was used in **G** and **H**.

The following figure supplements are available for figure 3:

**Figure supplement 1**. USP13 loss-of-function causes TCRα to accumulate in aggregates in cells.

To further characterize the localization of the TCRα-containing aggregates, we examined the localization of stabilized TCRα in cells expressing either USP13 or USP13 C345S. Compared to control cells in which TCRα was expressed at low levels and displayed a reticulum-like pattern co-localized with the ER marker PDI (*Figure 3—figure supplement 1B*, left column), ~50% of the USP13- and USP13 C345S-expressing cells contained TCRα-positive aggregates (*Figure 3—figure supplement 1B*, middle and right columns). These punctae were not co-localized with PDI, suggesting that they had left the ER. The fact that similar ERAD phenotypes were detected under both USP13 overexpression and depletion conditions further confirmed that USP13 overexpression inhibits ERAD by a dominant negative manner.

We also tested the effect of USP13 knockdown on the solubility of several other ERAD substrates including a transmembrane domain (TMD)-deleted TCRα variant predicted to be an ERAD-C substrate (*Soetandyo et al., 2010*), and the non-glycosylated ERAD substrate TTR D18G (*Christianson et al., 2011*). Cells expressing these ERAD substrates together with USP13 shRNA were subjected to extraction using the NP40 lysis buffer followed by the Laemmli buffer. Immunoblotting showed that USP13 depletion caused accumulation of these substrates in cells. Importantly, in both cases, more substrates were detected in the NP40-insoluble fractions in USP13 knockdown cells than in control cells (*Figure 3H,I*). Moreover, consistent with the model that USP13 acts together with Bag6 at a post-dislocation step, a significant fraction of TCRα-ΔTMD accumulated in USP13- and Bag6-knockdown cells in a deglycosylated form (*Figure 3H*, *Figure 3—figure supplement 1C*). These results demonstrate a function of USP13 in ERAD downstream of retrotranslocation and suggest a potential functional interplay between USP13 and Bag6 that is required to maintain retrotranslocated substrates in a soluble and degradable state.

## USP13 forms a multiprotein complex with gp78 and Bag6

We next tested whether USP13 could physically interact with Bag6 and/or its cofactors. Immunoprecipitation of USP13 from a whole cell extract resulted in co-precipitation of a fraction of Bag6 and its cofactor Ubl4A in addition to gp78 and p97. By contrast, the cytosolic chaperone Hsp90 was not co-precipitated (*Figure 4A*), demonstrating a specific interaction between USP13 and the Bag6 complex. The interaction of USP13 with the Bag6 complex and gp78 did not require its deubiquitinating activity (*Figure 4B*). This interaction could be further demonstrated by pull-down experiments using GST-tagged USP13 and whole cell extracts (*Figure 4C*). Notably, USP13 not only bound Bag6, but also interacted with MMS1 and α6, subunits of the proteasome (lane 5), further implicating USP13 in delivery pathways that targets substrates to the proteasome. Because depletion of gp78 did not affect the interaction of USP13 with Bag6 (*Figure 4C*, lane 7 vs 5), USP13 might bind Bag6 directly. Indeed, GST pull-down experiments using purified proteins showed that GST-USP13 bound Bag6, and the N-terminal ubiquitin-like (UBL) domain in Bag6 was both necessary and sufficient for interaction with USP13 (*Figure 4D,E*). Taken together, these results suggest a direct interaction between USP13 and Bag6 that is mediated by the Bag6 UBL domain.

Since Bag6 can interact with both gp78 and USP13, we asked whether these proteins could form a multiprotein complex. We performed sequential immunoprecipitation using cells expressing FLAG-USP13 and untagged gp78. As expected, proteins eluted from FLAG beads contained gp78 and each component of the Bag6 complex in addition to FLAG-USP13 (*Figure 4F*, lane 2). When this eluate was

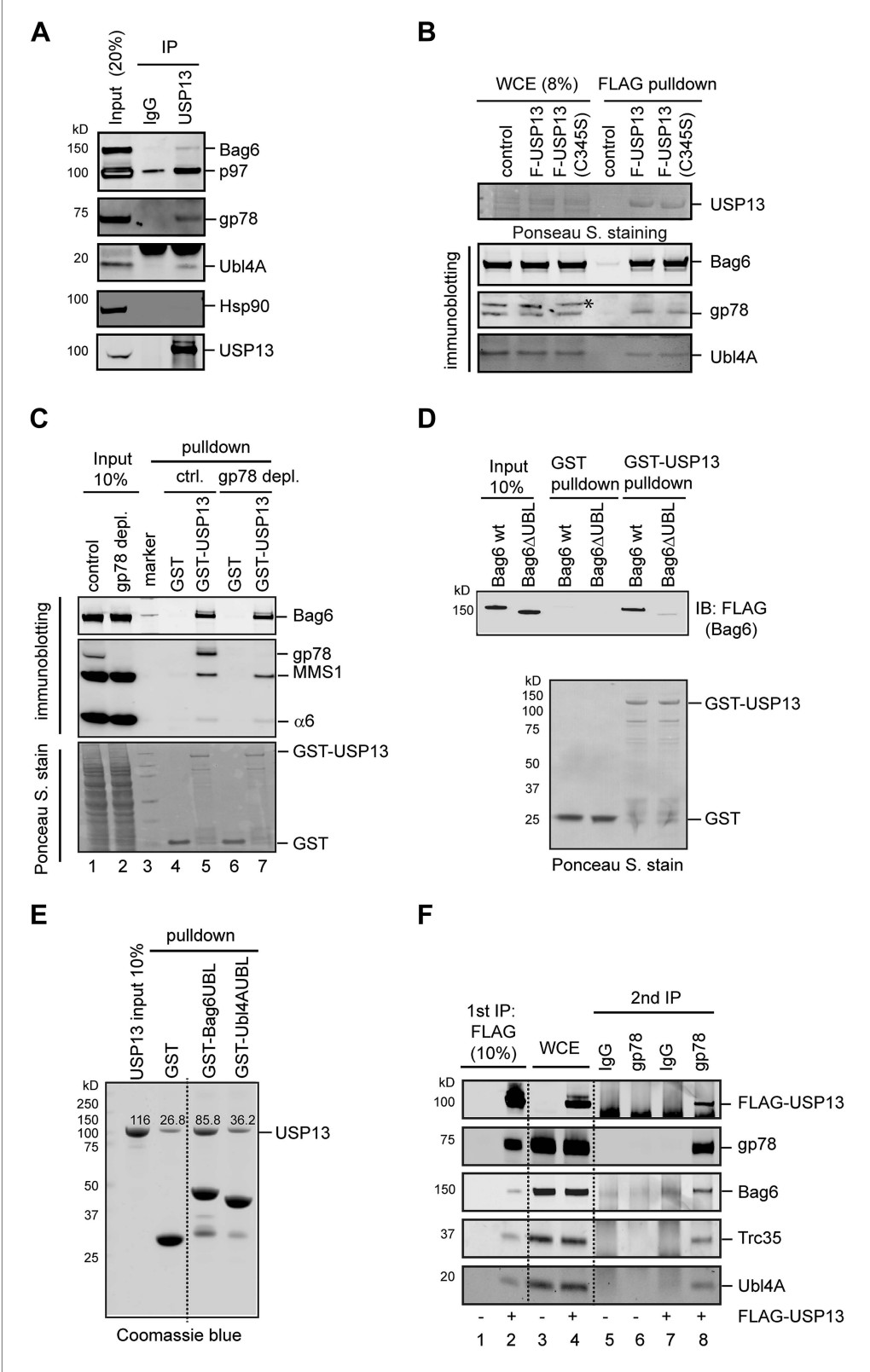

**Figure 4**. USP13 interacts with the Bag6 complex. (**A**) Endogenous interaction of USP13 with the Bag6 complex. Extracts from cells were subject to IP by the indicated antibodies followed by immunoblotting. (**B**) USP13 binds gp78 and Bag6 independent of its DUB activity. Extracts from HEK293 cells-transfected with the indicated plasmids

*Figure 4. Continued on next page*

*Figure 4. Continued*

were subject to IP with anti-FLAG beads. Asterisk indicates a non-specific band. (**C**) USP13 interacts with both the proteasome and Bag6 in a gp78 independent manner. Control (ctrl.) or gp78-depleted cell extracts were incubated with glutathione beads containing either GST or GST-USP13. The precipitated materials were analyzed by immunoblotting (Upper two panels) or by Ponceau S staining (the lower panel). (**D**) The UBL domain in Bag6 is required for interaction with USP13. A GST pull-down experiment was performed using the indicated Bag6 proteins purified from mammalian cells. (**E**) Bag6 UBL binds USP13 more tightly than Ubl4A UBL. Recombinant proteins purified from *E. coli* were used. (**F**) USP13, gp78 and Bag6 form a multi-protein complex. Sequential IP first by FLAG beads then by gp78 antibody using extracts from either control (−) or FLAG-USP13 (F-USP13)-expressing cells (indicated by '+').

---

subject to another round of immunoprecipitation using gp78 antibody, a significant fraction of the Bag6 complex remained bound to gp78 together with USP13 (lane 8), demonstrating that these proteins can form a multiprotein complex.

## USP13 and gp78 control ubiquitination of Ubl4A

Given that USP13 and gp78 are enzymes modulating ubiquitin dynamics, we reasoned that they might regulate Bag6 by controlling the ubiquitination status of a component in the Bag6 complex. We first asked whether Bag6 itself could be a substrate of USP13. Immunoblotting analyses of immuno-precipitated Bag6 failed to detect significant ubiquitin conjugates on Bag6 either in control or USP13 knockdown cells (*Figure 5—figure supplement 1A*), suggesting that Bag6 is unlikely regulated by USP13-mediated deubiquitination.

We next used the same approach to determine the ubiquitination status of Ubl4A and Trc35, the two Bag6 partners. Our results showed that Ubl4A, but not Trc35, could be ubiquitinated in cells (*Figure 5A*, lane 3 vs 7). Biochemical fractionation experiments showed that ubiquitination of Ubl4A preferentially occurred on the membrane-associated pool of Ubl4A (*Figure 5B*, lane 2 vs 1), consistent with the notion that it may be functionally relevant to ERAD regulation. Ubl4A appeared to be conjugated with Lys48-linked ubiquitin chains because expression of the ubiquitin K48R mutant, but not the K63R mutant, significantly reduced ubiquitinated Ubl4A (*Figure 5C*, lane 2 vs 1, 3). Intriguingly, despite carrying Lys48-linked ubiquitin chains, Ubl4A was a stable protein (*Figure 5—figure supplement 1B*, *Figure 3C*) ('Discussion').

If USP13 regulated Ubl4A ubiquitination, depletion of USP13 was expected to cause accumulation of ubiquitinated Ubl4A. Immunoblotting showed that Ubl4A purified from either USP13 knockdown cells or USP13 deficient cells indeed carried more ubiquitin conjugates than that from control cells (*Figure 5D,E*). Given that knockdown of the ER-localized deubiquitinase USP19 (*Hassink et al., 2009*) did not result in accumulation of ubiquitinated Ubl4A (*Figure 5F*), we concluded that ubiquitination of Ubl4A is specifically regulated by USP13.

Because USP13 depletion also led to accumulation of ubiquitinated ERAD substrate TCRα (*Figure 5—figure supplement 1C*, lane 2), it is important to distinguish whether USP13-mediated deubiquitination acts on Ubl4A, the ERAD substrates, or both. To this end, we screened several pH conditions in order to find a way to activate USP13 because previous studies showed that increased pH can facilitate depro-tonation of the catalytic cysteine in some USP DUBs to promote their activities (*Lee et al., 2013*). Indeed, at pH 8.0, the activity of USP13 was significantly elevated compared to pH 7.4 (*Figure 5—figure supplement 1D*). We then treated endogenous ubiquitinated Ubl4A purified from USP13-depleted cells with recombinant USP13 under this condition. This treatment did significantly reduce ubiquitinated Ubl4A, as demonstrated by immunoblotting (*Figure 5G*, lane 4 vs 1, *Figure 5—figure supplement 1E*), which was blocked by the specific DUB inhibitor ubiquitin aldehyde (*Figure 5G*, lanes 5–8 vs 1–4). As a control, we treated purified ERAD substrate TCRα under the same condition, but no obvious deubiquitination was observed (*Figure 5H*). Together, these results suggest that USP13 preferentially removes ubiquitin conjugates from Ubl4A, probably because it forms a specific interaction with the Bag6 complex via the Bag6 UBL domain (*Figure 4*). Although our results have not ruled out the possibility that USP13 may contribute to deubiquitination of TCRα with the assistance of a substrate recruiting adaptor in cells, the inability to deubiquitinate TCRα by purified USP13 suggests that the accumulation of ubiquitinated TCRα in USP13 knockdown cells is probably due to ERAD inhibition at a step downstream of ubiquitination, as shown in Bag6-depleted cells (*Wang et al., 2011*).

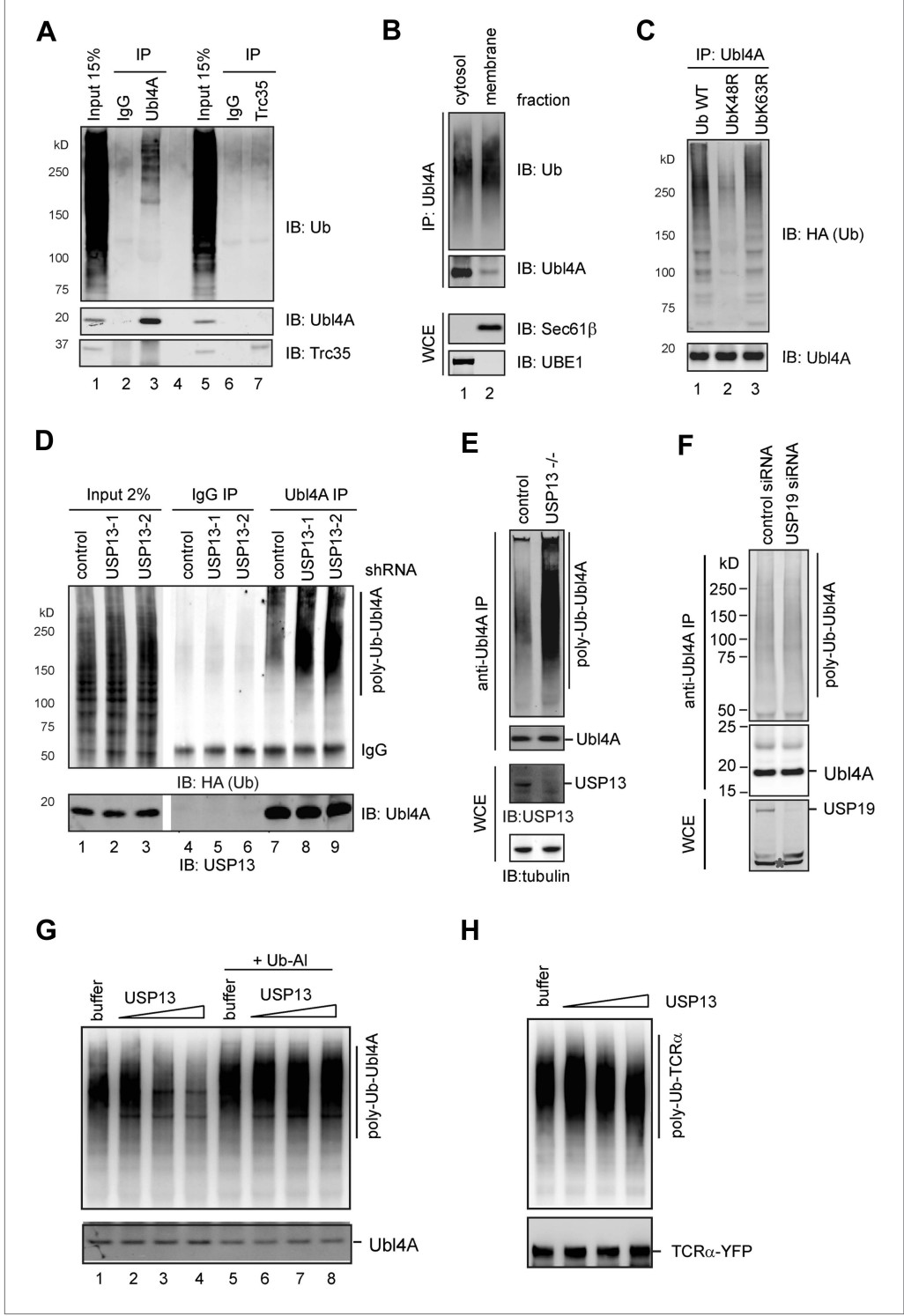

**Figure 5**. USP13 regulates ubiquitination of Ubl4A A, Ubl4A but not Trc35 is ubiquitinated in cells. (**A**) Ubl4A or Trc35 was immunoprecipitated from an extract of HEK293 cells under denaturing condition and blotted with the indicated antibodies. (**B**) Membrane-associated Ubl4A is preferentially ubiquitinated. HEK293 cells were fractionated into an ER-containing membrane and cytosol fractions. Ubl4A immunoprecipitated from these fractions was analyzed by immunoblotting. (**C**) Ubl4A is conjugated with Lys48-linked ubiquitin chains. Ubl4A immunoprecipitated from extracts of cells expressing the indicated ubiquitin variants was analyzed by immunoblotting. (**D** and **E**) USP13 depletion causes accumulation of ubiquitinated Ubl4A in cells. (**D**) Ubiquitinated Ubl4A was analyzed in cells

*Figure 5. Continued on next page*

*Figure 5. Continued*

expressing the indicated shRNAs and HA-ubiquitin. (**E**) Ubiquitinated Ubl4A was analyzed in control wild-type cells and USP13 knockout (−/−) cells expressing HA-ubiquitin. (**F**) USP19 depletion does not affect Ubl4A ubiquitination. Asterisk indicates a non-specific band. (**G**) In vitro deubiquitination of Ubl4A by USP13. Ubl4A purified from USP13 knockout cells was treated with either buffer or increased concentration of FLAG-USP13 (0.15 μM–0.6 μM) at 37°C for 2 hr. Where indicated, Ub-aldehyde (Ub-Al, 1 μM) was included. (**H**) TCRα purified from USP13 knockout cells was incubated with either buffer or FLAG-USP13 (0.15 μM–0.6 μM) at 37°C for 2 hr.

The following figure supplements are available for figure 5:

**Figure supplement 1**. Regulation of Ubl4A by ubiquitination.

Given that membrane-associated Ubl4A is preferentially ubiquitinated, we hypothesized that gp78 might contribute to Ubl4A ubiquitination. Incubating purified Ubl4A, E1, Ube2g2, ubiquitin with wild-type gp78c, but not a catalytically inactive gp78c mutant resulted in accumulation of high molecular weight products corresponding to ubiquitinated Ubl4A (*Figure 6A*, lane 2 vs 3). Likewise, overexpression of wild-type gp78, but not the catalytically inactive RING mutant (gp78 Rm) in cells significantly increased ubiquitination of endogenous Ubl4A (*Figure 6B*). These data suggest that gp78 can ubiquitinate Ubl4A directly in vitro and in vivo. Surprisingly, knockdown of gp78 did not significantly reduce ubiquitinated Ubl4A (*Figure 6C*). One possible way to explain these seemingly contradictory results is that Ubl4A might be ubiquitinated by several Bag6-associated ligases, but only those conjugates assembled by gp78 were preferentially removed by USP13. If this was correct, ubiquitinated Ubl4A accumulated upon USP13 depletion should be reduced when gp78 was depleted together with USP13, which was indeed observed in USP13 knockdown (*Figure 6D*, lane 9 vs 8) or deficient cells expressing gp78 specific shRNA (data not shown). These data suggest that USP13 and gp78 play antagonizing roles in regulation of Ubl4A ubiquitination: While gp78 assembles ubiquitin chains on Ubl4A, USP13 antagonizes this activity to limit Ubl4A ubiquitination.

## Hyper-ubiquitination of Ubl4A is associated with Bag6 clipping

To see how ubiquitination of Ubl4A influenced the function of the Bag6 complex, we analyzed the Bag6 protein level in USP13-depleted cells by immunoblotting. Intriguingly, we observed that a fraction of Bag6 (designated as Bag6*) had increased mobility on SDS-PAGE gel (*Figure 6D*, lane 5). This fast migrating Bag6 species was probably caused by proteolytic processing of Bag6 that removed a small C-terminal segment because Bag6* could not be detected by an antibody against the C-terminus of Bag6 (*Figure 3G*). Bag6* was primarily detected in NP40-insoluble fractions (*Figure 6D*, lane 5 vs 1), and its generation appears to be associated with hyper-ubiquitination of Ubl4A because co-depletion of gp78 together with USP13 reduced ubiquitinated Ubl4A and also abolished Bag6* accumulation (Lane 6 vs 5). Moreover, Bag6* was preferentially detected in the membrane bound pool of Bag6 in USP13-depleted cells (*Figure 7A*). Coincidentally, membrane-associated Ubl4A was more prone to ubiquitination (*Figure 5B*). Furthermore, overexpression of gp78, a condition that significantly increased ubiquitinated Ubl4A, also induced Bag6 cleavage (*Figure 7B,C*). Expression of a ubiquitin-Ubl4A fusion protein (Ub-Ubl4A) but not wild-type Ubl4A also caused accumulation of Bag6* in cells (*Figure 7D*). Importantly, even though overexpression of gp78 was previously shown to promote turnover of an ERAD substrate (CD3δ) (*Fang et al., 2001*) whose degradation is independent of Bag6 (data not shown), the degradation of the Bag6 substrate TCRα was significantly inhibited in cells expressing gp78 (*Figure 7E*). Moreover, expression of Ub-Ubl4A also significantly reduced the rate of TCRα degradation (*Figure 7F*). Together, these results suggest that hyper-ubiquitination of Ubl4A can cause ERAD inhibition in a dominant negative manner, probably because it is associated with the cleavage and subsequent inactivation of a fraction of membrane bound Bag6.

## Ubiquitination of Ubl4A preferentially occurs on Lys48

To further dissect the mechanism by which ubiquitination of Ubl4A inhibits Bag6 function, we used mass spectrometry to map the ubiquitination site in Ubl4A. The result identified Lys48 in Ubl4A as the major ubiquitination site (*Figure 8—figure supplement 1A*). To confirm that this Lys residue was ubiquitinated in cells, we initially attempted to overexpress a Ubl4A mutant bearing the Lys48 to Arg substitution. However, we found that most transiently expressed Ubl4A was not incorporated into the

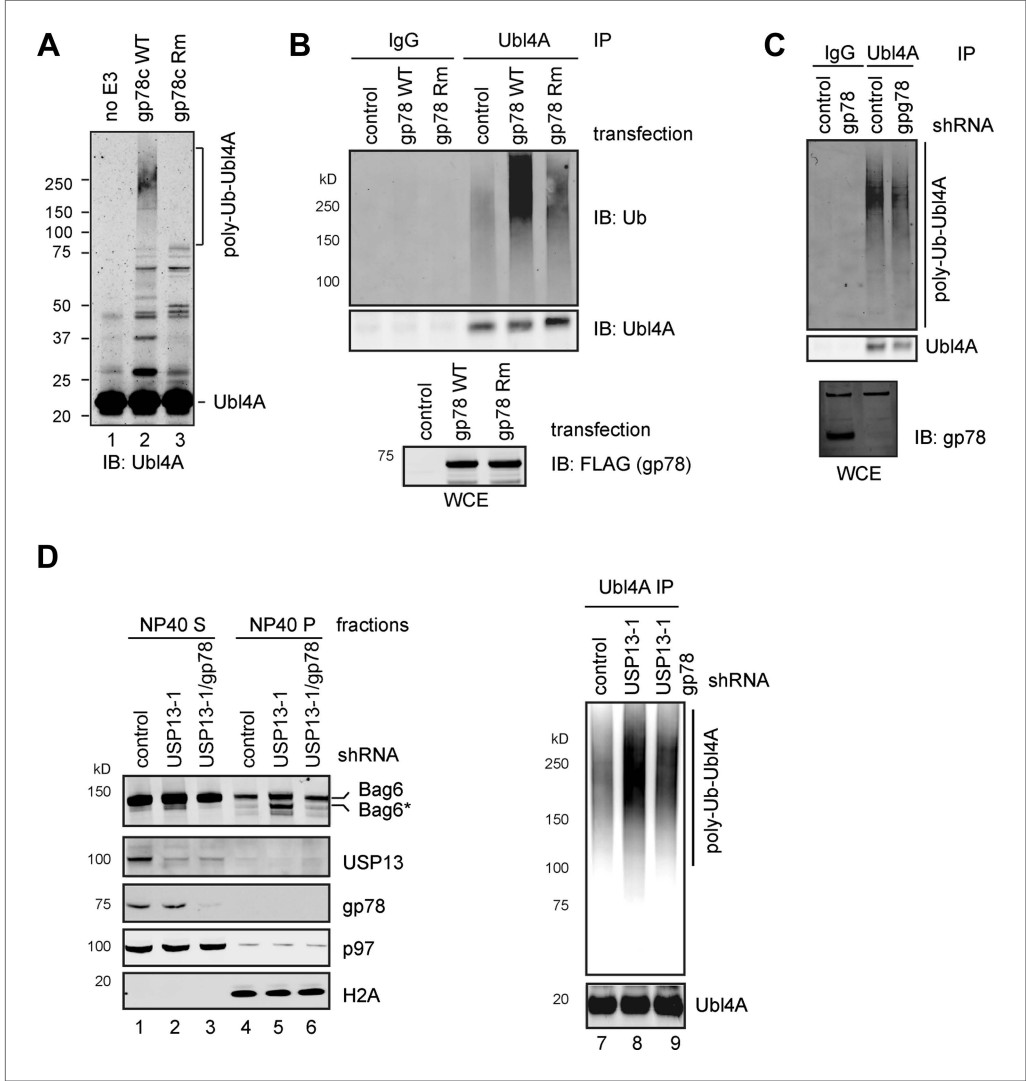

**Figure 6**. USP13 preferentially removes gp78-assembled ubiquitin chain from Ubl4A. (**A**) In vitro ubiquitination of Ubl4A by gp78. Purified Ubl4A was incubated with E1, Ube2g2, ubiquitin, ATP in the absence (no E3) or presence of the wild-type gp78 cytosolic domain (gp78c) or a catalytically inactive gp78c RING mutant (Rm). The reactions were analyzed by immunoblotting with anti-Ubl4A antibody. (**B**) Overexpression of wild-type gp78 promotes ubiquitination of Ubl4A. HEK293 cells were transfected with control or the indicated gp78-expressing plasmid. Ubiquitination of endogenous Ubl4A was analyzed by immunoprecipitation followed by immunoblotting. WCE, whole cell extracts. (**C**) gp78 knockdown in cells does not significantly reduce basal ubiquitination of Ubl4A. Ubl4A immunoprecipitated under denaturing conditions from extracts of cells transfected with the indicated shRNA constructs was analyzed by immunoblotting. Where indicated, a fraction of the whole cell extract was directly analyzed by immunoblotting to verify the knockdown efficiency. Asterisk indicates a non-specific band, which serves as a loading control. (**D**) gp78 antagonizes USP13 in regulation of Ubl4A. Lane 1–6, A fraction of the cells expressing the indicated shRNAs and HA-Ubiquitin were subject to sequential extraction by NP40- and SDS-containing buffers. Lane 7–9, the remaining cells were used to measure the levels of ubiquitinated Ubl4A as in **B**.

endogenous Bag6 complex because the complex had a slow turnover rate. As a result, overexpressed Ubl4A was not regulated by the same manner as endogenous Ubl4A. To overcome this problem, we established stable cell lines expressing either FLAG-tagged wild-type Ubl4A or Ubl4A K48R mutant under an inducible promoter. A careful titration experiment showed that when cells were grown in the presence of a low concentration of the inducer tetracycline ('Materials and method'), Ubl4A and Ubl4A K48R could be expressed at a level comparable to that of endogenous Ubl4A, and a significant fraction of ectopic Ubl4A interacted with Bag6. Under this condition, ubiquitination still preferentially occurred

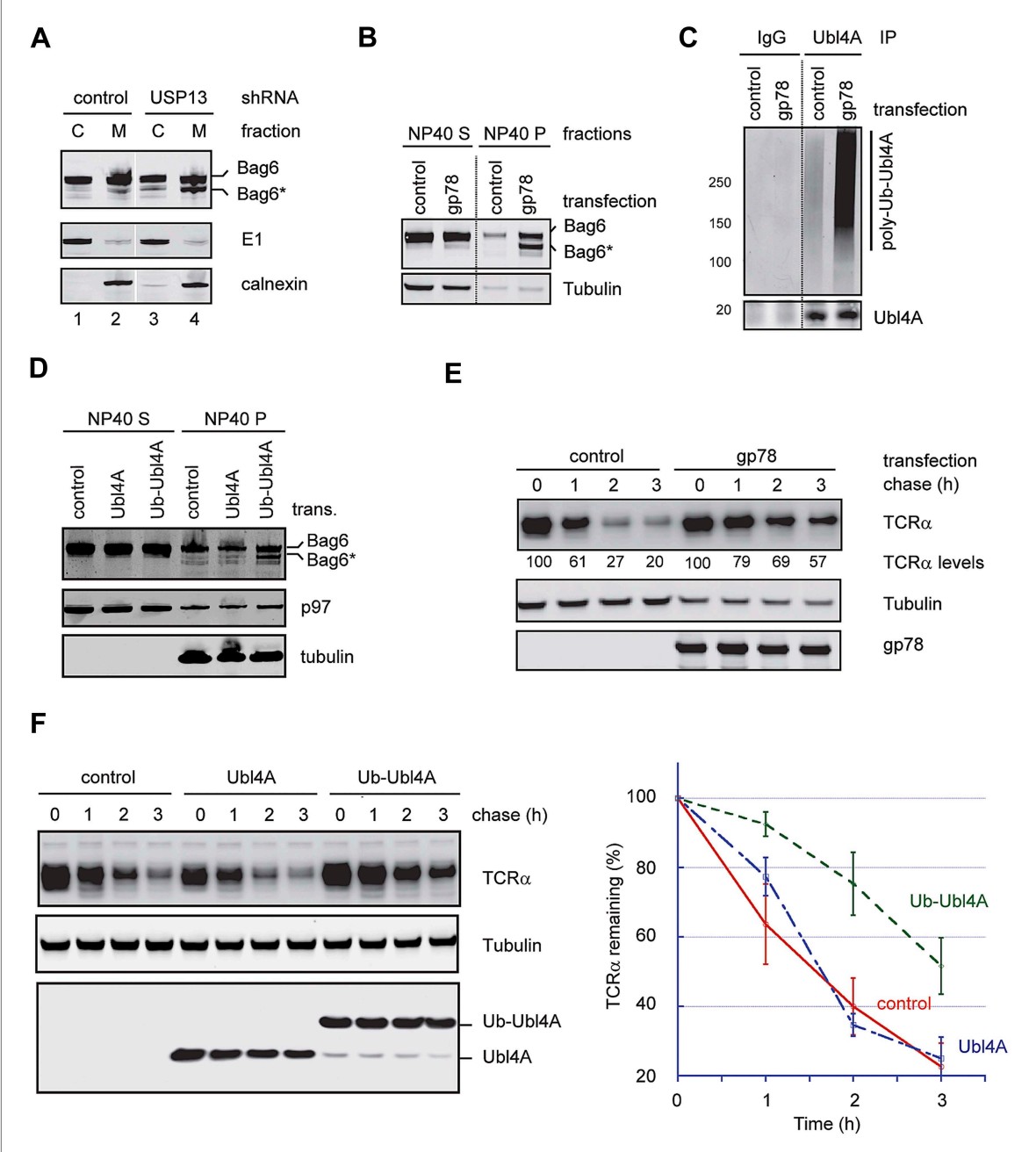

**Figure 7**. Hyper-ubiquitination of Ubl4A is associated with Bag6 clipping and ERAD inhibition. (**A**) Membrane-associated Bag6 is preferentially cleaved in USP13 knockdown cells. C, cytosol fraction; M, membrane fraction. (**B and C**) Overexpression of gp78 causes cleavage of Bag6. (**B**) A fraction of the cells transfected as indicated were extracted sequentially with NP40- and SDS-containing buffers. The corresponding extracts were analyzed by immunoblotting. (**C**) Protein extracts from HEK293 cells transfected with control or gp78-expressing plasmid were subject to immunoprecipitation with Ubl4A antibodies or with IgG as a negative control. (**D**) Overexpression of a Ub-Ubl4A fusion protein induces Bag6 cleavage. As in **B**, except that cells expressing the indicated proteins were analyzed. (**E**) Overexpression of wild-type gp78 inhibits TCRα degradation. The number indicates the relative levels of TCRα averaged from two independent experiments. (**F**) Overexpression of Ub-Ubl4A inhibits TCRα degradation. The graph shows the quantification results from three independent experiments. Error bars, SD (n = 3).

on endogenous Ubl4A (data not shown). However, significant amount of ubiquitinated Ubl4A-FLAG could be detected when gp78 was expressed (*Figure 8A*, lane 7). By contrast, overexpression of gp78 did not lead to ubiquitination of the Ubl4A K48R mutant (lane 8), suggesting that gp78-mediated ubiquitination primarily occurs at Lys48 in Ubl4A in cells. Moreover, when HA-tagged ubiquitin was

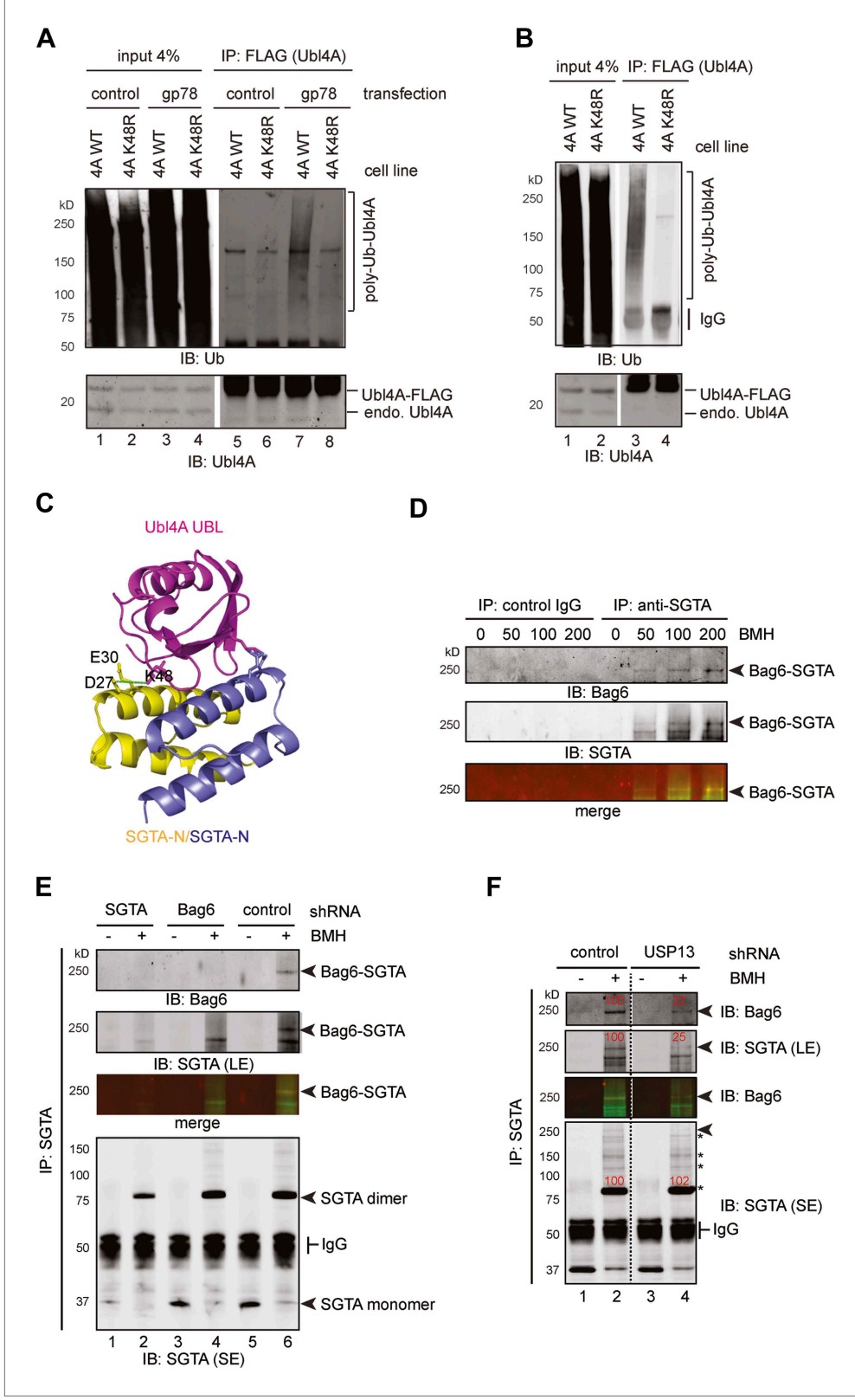

**Figure 8**. USP13 maintains a functional interaction between Bag6 and SGTA. (**A**) Lys48 of Ubl4A is required for gp78-mediated ubiquitination in cells. Cells stably expressing wild type and K48R mutant Ubl4A were transfected with either an empty vector (control) or a gp78-expressing vector. Whole cell extracts were either directly analyzed

*Figure 8. Continued on next page*

*Figure 8. Continued*

by immunoblotting (lanes 1–4) or subject to immunoprecipitation with anti-FLAG antibodies prior to immunoblotting (lane 5–8). (**B**) Mutating Lys48 in Ubl4A reduces basal ubiquitination of Ubl4A. Ubl4A ubiquitination was analyzed using wild type and the K48R mutant Ubl4A cells transfected with an HA-Ub expressing plasmid. (**C**) A model of SGTA-N-Ubl4A UBL complex. The structure of SGTA-N domain (PDB: 4GOD) (*Chartron et al., 2012*) and Ubl4A UBL domain (PDB: 2DZI) was superimposed on the complex of the corresponding yeast complex (PDB: 2LXC) (*Chartron et al., 2012*). Note that Lys48 in Ubl4A interacts with two negatively-charged residues in SGTA. (**D**) A Bag6- and SGTA-containing complex can be crosslinked in cells. HEK293 cells were treated with the indicated concentrations of BMH prior to lysis. Cell extracts were subject to IP with either IgG as a control or anti-SGTA antibody followed by immunoblotting. The arrow indicates a crosslinking species that is detected by both anti-Bag6 and anti-SGTA blotting. (**E**) As in **D**, except that cells expressing the indicated shRNA constructs were used and that IP was only conducted with the anti-SGTA antibody. SE, short exposure; LE, long exposure. (**F**) As in **E**, except that cells transfected with either control or USP13 shRNA were used. Asterisks indicate SGTA-containing crosslinking products that are not affected by USP13 knockdown. The numbers show the relative levels for the crosslinking products.
The following figure supplements are available for figure 8:

**Figure supplement 1**. Bag6 and SGTA form a transient interaction dependent on Ubl4A.

expressed in these cells, gp78-independent basal ubiquitination of Ubl4A could be detected on wild-type Ubl4A, but not on the Ubl4A K48R mutant (*Figure 8B*, lane 3 vs 4). These results establish Lys48 as the major site in Ubl4A for both gp78-dependent and gp78-independent ubiquitination.

## USP13 regulates an interaction of Bag6 with SGTA

Because our structural and biochemical studies showed previously that Ubl4A promotes Bag6 binding to a co-chaperone named SGTA via positively-charged residues including Lys48 on Ubl4A (*Figure 8C*) (*Chartron et al., 2012*; *Xu et al., 2012*), and because Lys48 is the major site of ubiquitination in Ubl4A, hyper-ubiquitination of Ubl4A under USP13 knockdown conditions might inhibit directly the interaction of Ubl4A with SGTA, and therefore disrupts this functional link between Bag6 and SGTA.

To follow the transient interaction between SGTA and the Bag6 complex, we employed a crosslinking approach. We treated HEK293 cells with a cysteine reactive crosslinker to stabilize protein–protein interactions. Cell extracts were subject to immunoprecipitation under denaturing conditions with SGTA antibody. Immunoblotting with SGTA antibodies detected a major crosslinking product whose molecular weight was consistent with a SGTA dimer. In addition, several SGTA-containing crosslinking products were detected only in anti-SGTA precipitated samples, but not in control samples (IgG pulldown). Among them, a crosslinking product of 250 kD was detected by both anti-SGTA and anti-Bag6 blotting (*Figure 8D*). Two pieces of evidence further suggested that this 250 kD crosslinking product was a crosslinked complex containing both SGTA and Bag6. First, in vitro crosslinking using purified SGTA and the Bag6 complex generated a similar 250 kD complex (*Figure 8—figure supplement 1B,C*). This complex was not detected when the Bag6 complex or SGTA was omitted from the reaction. Second, knockdown of either Bag6 or SGTA by shRNA significantly reduced this crosslinking product (*Figure 8E*, lanes 2, 4 vs 6). Consistent with the notion that Ubl4A facilitates the association of SGTA with Bag6, knockdown of Ubl4A also decreased this crosslinking product (*Figure 8—figure supplement 1D*). Importantly, USP13 knockdown specifically reduced the 250 kD Bag6-SGTA crosslinking product without affecting either the SGTA dimer or other SGTA-containing crosslinking products (*Figure 8F*). Collectively, these results indicate that USP13 facilitates the association between SGTA and Bag6.

## Discussion

We report a novel functional link between the deubiquitinase USP13 and gp78, an ER-associated E3 postulated to be part of a 'retrotranslocon' for some misfolded proteins (*Fang et al., 2001*). Despite having opposing activities, these enzymes interact with each other to form a complex that coordinately promotes ERAD. We identify the Bag6 cofactor Ubl4A as a shared substrate of gp78 and USP13. USP13 depletion is associated with hyper-ubiquitination of Ubl4A and altered interaction between the Bag6 complex and its co-chaperone SGTA. SGTA is an ortholog of Sgt2p, a chaperone implicated in biogenesis of tail-anchored ER proteins in *S. cerevisiae* (*Wang et al., 2010*). Like Bag6, SGTA also contains a chaperone-like activity that preferentially binds hydrophobic segments. We recently demonstrated that this activity facilitates substrate binding by Bag6 in ERAD (*Xu et al., 2012*). We also

showed that the interaction of Bag6 with SGTA is facilitated by Ubl4A as the latter binds directly to SGTA in a highly dynamic manner (*Chartron et al., 2012*). Because the interaction of Ubl4A with SGTA is mediated by positively-charged residues in Ubl4A including Lys48 (*Chartron et al., 2012*; *Xu et al., 2012*), which happens to be the major ubiquitination site, the simplest model to explain reduced Bag6-SGTA interaction in USP13 knockdown cells is that ubiquitin conjugates on Ubl4A sterically hinder SGTA binding. However, given that USP13 can also interact physically with Bag6, it is also possible that USP13 may serve an adaptor function to promote Bag6–SGTA interaction.

Hyper-ubiquitination of Ubl4A in USP13-depleted cells is also associated with increased proteolysis of Bag6, resulting in a truncated variant (Bag6*). Based on the molecular weight, the cleavage seems to occur within or near the C-terminal BAG domain. It has been shown that the Bag6 co-factor Ubl4A binds to a site near the BAG domain (*Xu et al., 2013*). Thus, it is possible that Ubl4A normally covers a protease site in Bag6 either by itself or by recruiting a Bag6 cofactor. Hyper-ubiquitination of Ubl4A in USP13 knockdown cells may alter its function, leading to increased cleavage of Bag6. In support of this idea, Bag6* also accumulates in cells depleted of Ubl4A (*Figure 8—figure supplement 1E*) or cells expressing the Ub-Ubl4A fusion protein (*Figure 7D*). Our results suggest that depletion of USP13 can impair Bag6 function via at least two ways. One is to cause its cleavage and the other is to inhibit its interaction with SGTA (*Figure 8—figure supplement 1F*). The two events may be linked as increased cleavage of Bag6 may also contribute to reduced interaction with SGTA. Although hyper-ubiquitination of Ubl4A is clearly detrimental to ERAD, it is noteworthy that our results do not exclude the possibility that transient ubiquitination of Ubl4A by gp78 at the site of retrotranslocation may also serve a positive role in ERAD.

Because the accumulation of Bag6* upon USP13 depletion is significantly reduced in cells co-depleted of USP13 and gp78 (*Figure 6D*), we propose that USP13 is required to antagonize a promiscuous activity of gp78 towards Ubl4A, which would otherwise impair the function of the Bag6 complex by altering its interaction pattern and/or increasing its cleavage by a cellular protease. In this model, a DUB can cooperate with an E3 ligase to enhance its substrate specificity. The specificity of a ubiquitination reaction has generally been thought to be controlled at the E3 ligase level. In proteasomal degradation pathways, many E3s appear to directly recognize substrates bearing degradation signals, leading to their ubiquitination. However, in the complex cellular environment, ubiquitin ligases often function in large protein complexes, meaning that in addition to substrates, many cellular proteins containing ubiquitin acceptor lysine residues are also in proximity to these enzymes. How these E3 cofactors evade ubiquitination is completely unknown. Our study suggests that cooperation between an E3 ligase and an associated DUB may provide a simple solution that sharpens substrate specificity for the ligase. It is conceivable that while acting on substrates, E3 ligases may also ubiquitinate other factors that function in proximity. Such undesired ubiquitination, even occurring at a low frequency, could cause significant damage overtime, particularly if it leads to irreversible inactivation of the modified proteins. Removal of these unwanted ubiquitination products by DUBs ensures that only desired ubiquitination signals are maintained in cells. This concept is in line with a recent study, showing that even non-specific DUB activities can amplify the specificity of a quality control E3, allowing discrimination between two misfolded substrates bearing subtle structural differences (*Zhang et al., 2013*).

Our study also suggests that Lys48-linked ubiquitination can have a non-destructive function, which is to regulate protein–protein interactions. Previous studies showed that cells can use ubiquitin to regulate protein–protein interaction in many ways. The most common strategy is to have ubiquitin serve as a 'signal' that recruits a downstream effector containing a ubiquitin-binding domain. By contrast, the regulation of Bag6 by the gp78-USP13-mediated ubiquitination cycle seems to represent another paradigm; Ubiquitination inhibits the recruitment of a co-factor. It is noteworthy that a similar ubiquitin-dependent disassembly of a protein complex involved in TGFβ signaling has been reported (*Dupont et al., 2012*; *Eichhorn et al., 2012*).

How Ubl4A escape proteasomal degradation despite carrying a bone fide proteasome degradation signal remains to be investigated. It is possible that ubiquitin conjugates on Ubl4A is bound by one of the many ERAD factors that carry ubiquitin-binding motifs (*Xu et al., 2012*). These interactions may prevent substrate recognition by the proteasome. Such a mechanism has been reported to maintain the stability of the ubiquitinated transcription factor Met4 in yeast (*Flick et al., 2006*).

Our model predicts that the USP13 activity needs to be tightly regulated both spatially and temporally in cells, a theme generally applicable to DUBs (*Reyes-Turcu et al., 2006*). Indeed, we found that purified

USP13 is inactive despite sequence homology to USP5, a constitutively active isopeptidase. Our data suggest that sequence variations in a UBA domain allow USP13 to acquire new interactors, which may activate it at the ER membrane. Since purified gp78 is insufficient to activate USP13 (Liu, Y unpublished results), further experiments using gp78-containing retrotranslocation complexes reconstituted in proteoliposome are required to elucidate how USP13 is regulated in the context of retrotranslocation.

In summary, our study demonstrates that a ubiquitin ligase and a DUB can act in concert to coordinate protein turnover at the ER by regulating the ubiquitination status of a degradation machinery factor. In addition, our findings reveal a mechanism by which an E3-DUB 'tag team' can enhance E3 specificity, a paradigm that may be applicable to other E3-DUB complexes.

## Materials and methods

### Cell lines and plasmids

The HEK293 cell line stably expressing TCRα-YFP was described previously (*Soetandyo et al., 2010*). Mammalian expression constructs for Ufd1, Npl4, Derlin-1, Ube2g2, VIMP, Hrd1, and gp78 were described previously (*Ye et al., 2001*, *2003*, *2004*; *Li et al., 2007*). Constructs for expression of UbxD8, ASPL, UbxD5, SAK1, FAF1, UbxD1, UbxD7, p47 were kindly provided by R Deshaies (Caltech, Pasadena, CA) (*Alexandru et al., 2008*). FLAG-USP13 and FLAG-USP5 expression plasmids were constructed by cloning the corresponding cDNA into the SalI and NotI sites of the pRK5-FLAG vector (*Li et al., 2008*). The USP5/USP13 chimera constructs were made by fusing USP5 and USP13 DNA fragments (as indicated in *Figure 1B*) using a 'sewing' PCR method. GST-USP13 and GST-USP5 were made by cloning the coding regions of USP13 or USP5 to the SalI and NotI sites in the pET42c(+) vector (Novagen/Merck, Germany). The pET42-Bag6 UBL and pET42-Ubl4A UBL constructs were described previously (*Xu et al., 2012*). Constructs for expression of Erasin and gp78 shRNA were provided by M Monteiro, and S Fang, respectively (University of Maryland, Baltimore, MD). Plasmids for expression of Bag6 and its truncated variants were generated by cloning the coding DNA fragments as indicated in *Figure 4* into the EcoRV and NotI sites in the pRK5-FLAG vector. Plasmid expressing FLAG-tagged Ubl4A and FLAG-tagged SGTA were purchased from Origene Technologies (Rockville, MD). To generate the gp78 RING mutant for mammalian expression, Cys356 and His361 in the gp78 RING domain were converted to Glycine and Alanine, respectively. The pGEX-gp78c Rm construct was described previously (*Li et al., 2007*). The constructs for expression of the various GST-gp78c truncation mutants were generated by a PCR-based mutagenesis approach using the pGEX-gp78c as the template and the TagMaster Site-Directed Mutagenesis kit from GM Biosciences (Frederick, MD). Other point mutations were generated by PCR based site-directed mutagenesis following a standard protocol. All mutations were confirmed by DNA sequencing. To construct pCMV-Ub-Ubl4A, ubiquitin-coding sequence bearing a G76V mutation was first cloned into pET28a using the NdeI and SalI sites. Ubl4A coding sequence was then inserted downstream of Ub-G76V using HindIII and XhoI sites. The whole Ub-G76V-Ubl4A fragment was then amplified by PCR and sub-cloned into pCMV6-ENTRY vector using the SgfI and MluI sites. For USP13 knockdown experiments, the targeting sequences are as follows:

 USP13-1 shRNA: cctgaatacttggtagtgcagataaagaa
 USP13-2 shRNA: gcgcatgtttaaggcctttgt

The knockdown constructs were purchased from Origene (Rockville, MD). shRNA constructs for depletion of Bag6 and Ubl4A were described previously (*Wang et al., 2011*). TransIT-293 (Mirus) was used for DNA plasmid transfection except for the gene knockdown experiments in which transfections were carried out using lipofectamine2000 (Invitrogen). The HEK293 cell lines stably expressing Ubl4A-FLAG or Ubl4AK48R-FLAG were made according to the Invitrogen Fln-In T-Rex system protocol. Briefly, Ubl4A-FLAG or Ubl4AK48R-FLAG fragments were sub-cloned into the HindIII and BamHI sites in the pCDNAa5/FRT/TO vector. The construct, together with a help vector pOG44, was then transfected into the Flp-In T-Rex expression cells. Positive cell clones resistant to hygromycin (125 μg/ml) and Blasticidin (15 μg/ml) were pooled together and maintained in the presence of a low dose of tetracycline (3.3 ng/ml) to induce the expression of Ubl4A.

gp78 knock out cells were generated using the CRISPR/Cas9 technology (*Wiedenheft et al., 2012*). The vector pX330 (*Cong et al., 2013*) for inserting the guide sequence was purchased from

Addgene. The following two oligoes (Forward primer, 5'-CACCgcccagcctccgcacctaca; Reverse primer, 3'-GcgggtcggaggcgtggatgtCAAA-5') were annealed and the resulting double strand DNA fragment was inserted into pX330 at the BbsI sites. The construct was then transfected into 293T cells following the standard protocol. 48 hr post-transfection, 50% of the cells were used to prepare genomic DNA for SURVOR assay to validate the cleavage of the target DNA. The remaining cells were cloned by infinite dilution. Immunoblotting was used to validate the positive knockout clones. USP13 knockout cells were generated by the same way. The two oligos are 5'-CACCgccgcccggcatgccgaaca and; 3'-GcggcgggccgtacggcttgtCAAA-5'.

## Antibodies, chemicals, and proteins

The antibody for USP13 was raised against recombinant GST-USP13 protein. Rabbit polyclonal antibodies to Ubl4A, Trc35, gp78, Ube2g2, Histone H2A, ATF4, and GFP were described previously (*Li et al., 2009*; *Wang et al., 2009*, *2011*). The anti-SGTA antibody was raised against the following peptide (CRSRTPSASNDDQQE-COOH) and affinity-purified using the same peptide conjugated to Sulfolink coupling resin (Thermo Scientific). Unless specified in the figure legends, Bag6 antibodies raised against a recombinant protein containing the N-terminal 249 amino acids were used for immunoblotting to detect the full length Bag6 and the cleaved Bag6. Other antibodies used are FLAG (M2) (Sigma, St. Louis, MO), p97 (Fitzgerald, Acton, MA), UBE1 (Sigma), UbxD8 (Proteintech Group, Chicago, IL), Myc (Santa Cruz, Dallas, TX), Trc35 (Novus Biologicals, Littleton, CO), ubiquitin (P4D1, Santa Cruz), Bag6 (Creative BioMart, Shirley, NY), PDI (1D3), Hsp90 (AC88) (Enzo Life Sciences, Farmingdale, NY), and MMS1 (BIOMOL/Enzo, Farmingdale, NY). Trc35 antibody from Novus Biological was used for immunoprecipitation. HRP-linked or fluorescence-labeled secondary antibodies (Rockland, Boyertown, PA) were used for immunoblotting detection. GST-E1, ubiquitin, ubiquitin-aldehyde, methylated ubiquitin, and ubiquitin K48R were purchased from Boston Biochem. MG132 was purchased from EMD Bioscience. EndoH was purchased from New England Biolab.

## Protein expression and purification

The purification of Bag6 and the Bag6-Ubl4A-Trc35 complex were described previously (*Wang et al., 2011*). The truncated Bag6 variants, SGTA, FLAG-USP13 and the USP13/5 chimeras were all purified using the same FLAG affinity chromatography procedure. Ube2g2 and GST-gp78c were purified from *E. coli* according to a previously described method (*Ye et al., 2003*). GST-USP13 and GST-USP5 were purified from *E. coli* according to a published protocol (*Russell and Wilkinson, 2005*). Proteins eluted from glutathione column (GE Healthscience, Piscataway, NY) or Ni-NTA beads (Qiagen, Valencia, CA) were further fractionated on a Superdex 200 HR (10/30) column in a buffer containing 50 mM Tris-HCl, pH 8.0, 150 mM potassium chloride, 5% glycerol, 1 mM DTT, and 2 mM magnesium chloride.

## Immunoprecipitations, GST pulldown, and immunoblotting

Cell lysates were prepared using the NP40 lysis buffer (50 mM Tris–HCl pH 7.4, 150 mM sodium chloride, 2 mM magnesium chloride, 0.5% NP40, and a protease inhibitor cocktail). For most experiments, NP40 soluble fractions were used for immunoprecipitation or immunoblotting. Where indicated, the NP40-insoluble pellets were re-solubilized by the Laemmli buffer for immunoblotting. For immunoprecipitation under denaturing condition, cells were first lysed in a buffer containing 1%SDS and 5 mM DTT. The lysates were heated at 65°C for 15 min and then diluted 10-fold by the NP40 lysis buffer. The samples were subject to centrifugation at 20,000×*g* for 10 min to remove insoluble materials. The soluble supernatant fractions were used for immunoprecipitation by antibodies as indicated in the figure legends. For sequential immunoprecipitation, cell extract was first incubated with FLAG beads. The bound materials were eluted by a buffer containing FLAG peptide (0.2 mg/ml). The eluate was subject to another round of immunoprecipitation by anti-gp78 antibody or a control antibody. Immunoblotting was performed following the standard protocol.

## Flow cytometry and immunofluorescence microscopy

To measure the levels of YFP-tagged proteasomal substrates, cells were analyzed by the Cytomics FC 500MPL Flow Cytometry System (Beckman Coulter, Brea, CA) following a previously described procedure (*Wang et al., 2011*). For the immunofluorescence imaging studies, cells transiently expressing YFP-tagged TCRα were grown in a Lab-Tek slide chamber (Nalge Nunc International, Rochester, NY) were fixed for 20 min in phosphate-buffered saline (PBS) containing 4% paraformaldehyde. Images

were taken using an Axiovert 200 inverted microscope with a 63× oil objective lens (Zeiss, Germany). For immunostaining experiments, fixed cells were permeabilized in PBS containing 0.1% NP40 and 5% normal donkey serum and then stained in the same buffer with the indicated antibodies.

## In vitro ubiquitination and deubiquitination assays

Ubiquitination experiments were performed as described previously (Ye et al., 2003). Briefly, E1 (60 nM), Ube2g2 (200 nM), gp78c (400 nM), and ubiquitin (10 µM) were incubated with the substrate Ubl4A at 37°C for 1 hr in a buffer containing 25 mM Tris-HCl, pH 7.4, 2 mM magnesium/ATP, and 0.1 mM DTT. The DUB activities were measured by detecting the increase in fluorescence upon cleavage of Ubiquitin-AFC (Boston Biochem, MA). Purified DUBs (~50 nM) were added individually to 200 µl deubiquitinating assay buffer (50 mM Tris-HCl pH 7.4, 20 mM potassium chloride, 5 mM magnesium chloride, 1 mM DTT, including 0.5 µM Ub-AFC), and incubated at 37°C. The fluorescence intensity was measured using an Aminco Bowman Luminescence spectrometer with excitation and emission wavelengths set at 400 and 505 nm, respectively. In vitro deubiquitination of Ubl4A was performed as follows. USP13-depleted cells were transfected with HA-Ub and TCRα-YFP and lysed in the NP40 lysis buffer. The whole cell extract was incubated with protein A beads pre-bound with GFP or Ubl4A antibodies to purify TCRα and endogenous Ubl4A, respectively. After immunoprecipitation, the beads were washed twice with the NP40 washing buffer and once with 2× deubiquitination buffer. The beads were then divided into equal portions, each containing 20 µl 2× DUB buffer (100 mM Tris pH 8.0, 40 mM KCl, 10 mM MgCl$_2$, 2 mM DTT). The reactions were started by adding recombinant FLAG-USP13 (purified from mammalian cells) or GST-USP13 (purified from E. coli) as indicated in the figure legends. The samples were incubated at 37°C for 2 hr with shaking and then stopped by addition of 40 µl Laemmli buffer. The samples were then analyzed by immunoblotting with anti-HA (ubiquitin), anti-Ubl4A, or anti-GFP antibodies.

## Miscellaneous biochemical assays

In vivo crosslinking experiments were performed by incubating $2 \times 10^6$ cells in a labeling buffer (95 mM NaCl, 50 mM KCl, 10 mM glucose, 10 mM HEPES pH 7.2, 1.2 mM CaCl2, 1.2 mM MgCl2) containing the indicated concentration of 1,6-bis-Maleimidohexane (BMH) (Pierce) at room temperature for 20 min. The cells were then treated with 5 mM DTT for 10 min on ice, followed by addition of 1% SDS. Cell extracts were heated at 65°C for 15 min. The soluble extracts were diluted in the NP40 lysis buffer prior to immunoprecipitation using 2.5 µg affinity purified anti-SGTA antibody. In vitro crosslinking was performed similarly except that purified proteins were used. Cycloheximide chase experiments were performed by incubating cells in DMEM medium containing 50 µg/ml cycloheximide at 37°C. At the time points indicated in the figures, $\sim0.6 \times 10^6$ cells were harvested. The cell extracts were analyzed by immunoblotting.

## Acknowledgements

We thank R Deshaies (Caltech), S Fang and M Monteiro (University of Maryland) for plasmids, T Rapoport (Harvard Medical School) for anti-Sec61β antibodies, and T Rapoport and W Prinz for critical reading of the manuscript.

## Additional information

### Funding

| Funder | Author |
| --- | --- |
| National Institutes of Health Intramural Research Program | William M Clemons Jr, Yihong Ye |

The funder had no role in study design, data collection and interpretation, or the decision to submit the work for publication.

### Author contributions

YL, YY, Conception and design, Acquisition of data, Analysis and interpretation of data, Drafting or revising the article; NS, JL, LL, YX, Acquisition of data, Analysis and interpretation of data; WMC, Conception and design, Drafting or revising the article, Contributed unpublished essential data or reagents

## Additional files

### Major dataset

The following previously published datasets were used:

| Author(s) | Year | Dataset title | Dataset ID and/or URL | Database, license, and accessibility information |
|---|---|---|---|---|
| Chartron JW, Vandervelde DG, Clemons WM | 2012 | Solution structure of the complex between the Sgt2 homodimerization domain and the Get5 UBL domain | 2LXC; http://www.pdb.org/pdb/explore/explore.do?structureId=2LXC | Publicly available at RCSB Protein Data Bank (http://www.rcsb.org). |
| Chartron JW, Vandervelde DG, Clemons WM | 2012 | Crystal structure of the SGTA homodimerization domain | 4GOD; http://www.pdb.org/pdb/explore/explore.do?structureId=4GOD | Publicly available at RCSB Protein Data Bank (http://www.rcsb.org). |
| Zhao C, Sato M, Koshiba S, Watanabe S, Harada T, Kigawa T, Yokoyama S | 2009 | Solution Structure of the N-terminal Ubiquitin-like Domain in Human Ubiquitin-like Protein 4A (GDX) | 2DZI; http://www.pdb.org/pdb/explore/explore.do?structureId=2DZI | Publicly available at RCSB Protein Data Bank (http://www.rcsb.org). |

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
