## [Decision Letter]

Thank you for sending your work entitled “USP13 antagonizes gp78 to maintain functionality of a chaperone in ER-associated degradation” for consideration at *eLife*. Your article has been favorably evaluated by Randy Schekman as the Reviewing editor and two expert peer reviewers.

The Reviewing editor and the two reviewers discussed their comments before we reached this decision, and the Reviewing editor has assembled the following comments to help you prepare a revised submission.

ER-associated degradation is an important quality control mechanism that ensures the proteolytic removal of misfolded ER-proteins. The ER-associated ubiquitin ligase gp78 is a well-established, important regulator of ERAD that binds to the ubiquitin-selective chaperone p97. P97 also interacts with a series of different deubiquitinases (DUBs), including Usp13. Because ubiquitylation is required for proteasomal targeting, it is was a surprising finding that either loss of ubiquitin ligases (i.e., leading to decreased ubiquitylation) or loss of DUBs (i.e., leading to increased ubiquitylation) interfered with efficient ERAD. The reason for this paradox was unclear.

In this study, Liu et al. provide a potential answer to this paradox, i.e., in that they suggest that DUBs positively regulate ERAD by ensuring that functionality of ERAD-regulators that are complex with ubiquitin ligases. Having identified the ubiquitin ligase gp78 as potential bridging factor between the DUB Usp13 and p97, they show that Usp13 functions at a similar stage as the Bag6 chaperone complex recently isolated by the same group. Indeed, they provide very nice and convincing evidence that Usp13 acts in the same complex as Bag6. Next, they searched for substrates of Usp13 and identified a component of the Bag6-complex, Ubl4A, as a candidate. They suggest that Usp13 protects Ubl4A from spurious gp78-dependent ubiquitylation, thereby ensuring the Ubl4A can bind to downstream partners, such as SGTA. This part of the story, i.e., whether deubiquitylation of a fraction of Ubl4A-molecules in cells is required for the function of the Bag6-complex in vivo and whether this reaction is the key function of Usp13, is less well developed. Overall, especially the experiments in the first part of the paper are very well performed and interpreted; moreover, addressing the paradox of a dual requirement for DUBs and ubiquitin ligases is very important for the field. Thus, if the physiological relevance of the Usp13/Ubl4A axis can be strengthened, this paper is a candidate for publication by *eLife*.

1) One major weakness in this paper is the lack of direct evidence to demonstrate that USP13 removes the ubiquitin conjugates on Ubl4A. Obviously the most compelling data would be to show that purified recombinant USP13 can deubiquitinate the ubiquitin conjugates attached to Ubl4A. Given that USP13 did not display any deubiquitination activity in vitro (Figure 1), is this experiment possible? The closest data to support the authors' view are presented in Figure 5, in which USP13 knockdown increased Ubl4A polyubiquitination. Whether this is due to the lack of USP13 or other indirect consequences of silencing USP13 is unclear. Can the authors rescue this phenotype by using WT but not catalytic-inactive USP13? Because this is a critical result, a positive control in which another irrelevant DUB is silenced (e.g., USP5) should be performed. Additionally, as catalytic-inactive DUBs often trap their substrates, can a catalytic-inactive USP13 bind preferentially to Ubl4A?

2) More key data to support this model are presented in Figure 7 where USP13 knockdown disrupts the SGTA-Bag6 complex. However, because USP13 was shown to interact with the Bag6 complex in this paper, it is entirely possible that USP13 is simply acting as a scaffold to support the SGTA-Ubl4A-Bag6 complex. If so, USP13's catalytic function would not play any role in facilitating Bag6 complex formation. Again, can exogenous expression of WT but not catalytic-inactive USP13 rescue the formation of the SGTA-Bag6 complex? (Although the authors did demonstrate that WT USP13 over-expression can exert a dominant-negative phenotype during retro-translocation (Figure 2), this experiment was not performed in the context of silencing endogenous USP13.) Moreover, in this experiment, a double knockdown of USP13 and gp78 is expected to restore the SGTA-Bag6 complex. This finding would indicate that the lack of gp78-catalyzed Ubl4A ubiquitination negates the requirement of USP13 to deubiquitinate Ubl4A. The authors should do this experiment.

3) From a mechanistic point of view, it is unclear why Usp13 would target Ubl4A, but not the substrate (i.e., an event that would inhibit ERAD). If the model provided by the authors is correct, then one would expect that the effects of Usp13 depletion of ERAD-degradation (such as those seen in Figure 3) are rescued by expression of a Ubl4A-K48R mutant. Providing such evidence would immediately address the issue of physiological relevance in a very convincing manner.

4) The direct link between gp78 and Usp13 is fairly weak and to a large extent depends on the analysis of Usp5/Usp13 chimera. In this respect, the statement that the second UBA-domain accounts for the differential activity between Usp5 and Usp13 is not really supported by evidence (this would need a more specific swap of UBA-domains, rather than large protein segments). Adding these data would clearly strengthen the link between gp78 and Usp13. Moreover, the effect of gp78-depletion on the Usp13/p97 interaction is very weak and only one shRNA against gp78 is shown; this experiment needs to be repeated with multiple shRNAs (or better, rescue constructs), allowing the researchers to obtain a potential correlation between strength of knockdown and phenotype.

---

## [Author Response]

*[…] Overall, especially the experiments in the first part of the paper are very well performed and interpreted; moreover, addressing the paradox of a dual requirement for DUBs and ubiquitin ligases is very important for the field. Thus, if the physiological relevance of the Usp13/Ubl4A axis can be strengthened, this paper is a candidate for publication by* eLife*.*

We thank the reviewers for recognizing the importance of our work. We agree with the reviewers that the second part of our study, which links hyper-ubiquitination of the ERAD machinery protein Ubl4A to impaired Bag6 function is less developed. In fact, in our original manuscript, we described that a fraction of Bag6 undergoes a cleavage process to generate what we called Bag6* in USP13 depleted cells (Figure 6). The characterization of this cleaved product is an ongoing project that we initially wished to develop into an independent story. In light of the reviewers’ suggestions, we decide to include some results we obtained while characterizing this Bag6 cleavage product (Figure 7). We hope that this will help explain why hyper-ubiquitination of Ubl4A is detrimental to ERAD.

Specifically, we show that hyper-ubiquitination of Ubl4A, either as a result of lack of USP13 (Figure 6, lane 5; Figure 7) or increased gp78 (E3) ligase activity (Figure 7), is associated with increased cleavage of Bag6 by a yet-to-be identified cellular protease. This results in the accumulation of a truncated Bag6 variant lacking the essential BAG domain at the C-terminus. We believe that these results provide strong evidence to support our conclusion that the function of Bag6 is impaired when Ubl4A accumulates in ubiquitinated form. Importantly, when gp78 is co-depleted with USP13, it rescues Bag6 cleavage caused by USP13 depletion (Figure 6, lane 6 vs 5), strongly indicating that the two enzymes are functionally antagonizing each other to regulate Bag6. We initially wished to publish these results after we identify the cellular enzyme(s) responsible for this proteolysis reaction and map the cleavage site. Since the reviewers question how ubiquitination of Ubl4A can lead to inactivation of Bag6 without involving proteasome dependent turnover of the Bag6 complex, we hope that the inclusion of these results will to some extent alleviate this concern. We hope that the reviewers will agree that further characterization of this cleavage reaction is beyond the scope of the current study.

*1) One major weakness in this paper is the lack of direct evidence to demonstrate that USP13 removes the ubiquitin conjugates on Ubl4A. Obviously the most compelling data would be to show that purified recombinant USP13 can deubiquitinate the ubiquitin conjugates attached to Ubl4A. Given that USP13 did not display any deubiquitination activity in vitro (*Figure 1*), is this experiment possible? The closest data to support the authors' view are presented in*
Figure 5*, in which USP13 knockdown increased Ubl4A polyubiquitination. Whether this is due to the lack of USP13 or other indirect consequences of silencing USP13 is unclear. Can the authors rescue this phenotype by using WT but not catalytic-inactive USP13? Because this is a critical result, a positive control in which another irrelevant DUB is silenced (e.g., USP5) should be performed. Additionally, as catalytic-inactive DUBs often trap their substrates, can a catalytic-inactive USP13 bind preferentially to Ubl4A*?

To further support our conclusion that USP13 acts specifically on Ubl4A, we include an additional control experiment in which we silenced another deubiquitinase in cells. In this case, we chose USP19, which is the only DUB that contains a C-terminal transmembrane segment. This DUB has been previously implicated in ERAD. Another postdoc fellow in the lab has been working on this DUB for a while. We initially thought that it might perform a similar function as USP13 to regulate Bag6 in conjunction with another ER E3 ligase. However, when he depleted USP19 from the cells, it did not alter the level of ubiquitinated Ubl4A (Figure 5). The reviewers’ comments reminded us of this experiment that we did a while ago. Although it did not reveal what exactly USP19 is doing in ERAD, it provides the best control, showing that the effect of USP13 knockdown on Ubl4A ubiquitination is specific.

We also tried the in vitro deubiquitination experiments suggested by the reviewers. To overcome the issue of lack of activity, we screened several pH conditions as previous studies from us and others showed that increasing pH can often activate deubiquitinases because increased pH facilitates cysteine deprotonation. We find that purified USP13 indeed gains some activity at pH 8.0 compared to pH 7.4 (Figure 5—figure supplement 1). We then treated purified ubiquitinated Ubl4A with different doses of USP13 under pH 8.0 condition. Our data show that purified USP13 indeed can remove ubiquitin conjugates from Ubl4A (Figure 5; Figure 5—figure supplement 1). As anticipated, the deubquitination reaction can be inhibited by a DUB specific inhibitor ubiquitin aldehyde (Figure 5). Intriguingly, in a parallel experiment, when we treated the purified ERAD substrate TCRα with USP13, we did not observe significant deubiquitination (Figure 5). These results further support our model that USP13 acts directly on Ubl4A, but not on ERAD substrates, to influence ERAD.

Lastly, with regard to whether the catalytic inactive USP13 can trap Ubl4A, we actually showed in our original Figure 4 that the mutant USP13 does not bind Ubl4A better than wild type protein. The reviewers are absolutely right that some catalytically inactive DUB mutants can trap their substrates when overexpressed. Most of these cases involve USP DUBs that binds substrate primarily through ubiquitin conjugates. However, in the case of USP13, our results suggest that it does not contain high affinity binding site for ubiquitin (Figure 1—figure supplement 1). On the other hand, USP13 can bind a UBL domain in Bag6 (Figure 4), which explains why the catalytically inactive USP13 binds the Bag6-Ubl4A complex with similar affinity as the wild type protein.

*2) More key data to support this model are presented in*
Figure 7
*where USP13 knockdown disrupts the SGTA-Bag6 complex. However, because USP13 was shown to interact with the Bag6 complex in this paper, it is entirely possible that USP13 is simply acting as a scaffold to support the SGTA-Ubl4A-Bag6 complex. If so, USP13's catalytic function would not play any role in facilitating Bag6 complex formation. Again, can exogenous expression of WT but not catalytic-inactive USP13 rescue the formation of the SGTA-Bag6 complex? (Although the authors did demonstrate that WT USP13 over-expression can exert a dominant-negative phenotype during retro-translocation (*Figure 2*/G), this experiment was not performed in the context of silencing endogenous USP13.) Moreover, in this experiment, a double knockdown of USP13 and gp78 is expected to restore the SGTA-Bag6 complex. This finding would indicate that the lack of gp78-catalyzed Ubl4A ubiquitination negates the requirement of USP13 to deubiquitinate Ubl4A. The authors should do this experiment*.

We agree with the reviewers that the physical interaction of USP13 with the Bag6 complex makes it hard to distinguish whether the effect of USP13 knockdown on Bag6-SGTA interaction is due to lack of deubiquitination by USP13 or its scaffolding function. The proposed rescuing experiments are important. In fact, we have thought about doing this experiment and have spent a lot of time creating the necessary reagent. Specifically, we were able to generate a USP13 knockout cell line using the CRISPR technology recently. We have repeated a few assays shown in the paper with the knockout cells, which essentially gave rise to similar results as the knockdown cells. For example, ubiquitination of Ubl4A is significantly increased in the knockout cells compared to wild type control (Figure 5), which can be partially suppressed by depletion of the E3 gp78 (see Figure 9).Author response image 1.Author response image 1: Ubl4A ubiquitination was analyzed using control or USP13 knockout cells expressing either control or gp78 shRNA.

However, when we expressed wild type USP13 in this cell line, it still caused a dominant negative effect, resulting in significant increase in TCRα level (see Figure 10).Author response image 2.Author response image 2: USP13 knockout cells transfected with control or USP13 expressing plasmid together with TCRa-YFP were imaged using a Zeiss fluorescence microscope equipped with a 20X objective. The images were taken with the same exposure.

This result is consistent with our model that ERAD inhibition by USP13 overexpression is probably achieved by blocking functional protein–protein interactions required for ERAD through the binding of USP13 with gp78. It is noteworthy that dominant negative inhibition of cellular process by expression of wild type DUB is quite common, which makes the rescue experiment very tricky to do as we need to figure out a way to reproducibly express just the right amount of USP13 in order to avoid the dominant negative effect. Even making stable cell lines with retrovirus may not be sufficient because the expression of a transgene from a viral promoter is still going to be significantly higher than the endogenous one.

Although we cannot perform the suggested rescuing experiments, we feel that our new results showing that hyper-ubiquitination of Ubl4A is associated with increased Bag6 cleavage should offer a satisfying explanation as to how the functional integrity of the Bag6 complex is affected by USP13 depletion.

We also tried the proposed experiment to see if knockdown of gp78 could rescue the interaction between Bag6 and SGTA. To our surprise, we find that knockdown of gp78 just by itself (without USP13 knockdown) reduces the interaction of Bag6 with SGTA, as demonstrated by a crosslinking experiment (see Figure 11).Author response image 3.Author response image 3: Cells expressing either control or gp78 shRNA were treated with BMH. SGTA-containing complex was immunoprecipitated and analyzed by immunoblotting. A fraction of the whole cell extract was also analyzed. Note that the 250kD Bag6-SGTA crosslinking product was abolished in gp78 knockdown cells.

Because gp78 can interact with Bag6 directly, the result suggests that gp78 may serve a scaffolding function that promotes the interaction of Bag6 with SGTA. Importantly, the experiment also suggests that the crosslinked Bag6-SGTA complex is a result of an interaction that occurs in proximity to gp78. This result is consistent with our findings that ubiquitinated Ubl4A (Figure 5) and the Bag6 cleavage product (Figure 7) are all preferentially enriched in the ER membrane fractions. Although the correlations between these molecular events do not necessarily prove any causal relationships among them, our results suggest that hyper-ubiquitination of Ubl4A and cleavage of Bag6 are tightly coupled because expression of Ub-Ubl4A also induces Bag6 cleavage. Collectively, we think that our new data suggest two possible mechanisms by which depletion of USP13 can interfere with Bag6 function. One is to increase cleavage of Bag6 and the other is to cause reduced interaction with SGTA. Whether these two events are mechanistically linked awaits further study, but the results clearly demonstrate that USP13 participates in ERAD by an indirect mechanism that maintains the functionality of Bag6.

Because we cannot formally rule out the possibility that USP13 may also regulate Bag6 function by serving as a scaffold between Bag6 and SGTA, we now tone down our conclusion and mention this possibility in our paper. We also mention the possibility that increased Bag6 cleavage may also contribute to inhibition of Bag6 and SGTA interaction.

*3) From a mechanistic point of view, it is unclear why Usp13 would target Ubl4A, but not the substrate (i.e., an event that would inhibit ERAD). If the model provided by the authors is correct, then one would expect that the effects of Usp13 depletion of ERAD-degradation (such as those seen in*
Figure 3*) are rescued by expression of a Ubl4A-K48R mutant. Providing such evidence would immediately address the issue of physiological relevance in a very convincing manner*.

Our in vitro binding experiments showed that unlike the homologous DUB USP5, USP13 does not bind ubiquitin with a high affinity (Figure 1—figure supplement 1). On the other hand, the binding of USP13 with the UBL domain of Bag6 can be demonstrated using pulldown experiments (Figure 4). These experiments suggest that the interaction of USP13 with the substrate (the Bag6 complex) is distinct from the canonical DUB-substrate interactions, which are largely mediated by the ubiquitin conjugates on substrates. This provides a plausible explanation as to why USP13 won’t just act on any ubiquitinated proteins. We agree with the reviewers that rescuing the ERAD phenotype with a Ubl4AK48R mutant will provide valuable information on whether ubiquitination of Ubl4A on Lys48 is the only USP13 depletion-associated deficiency that causes ERAD defects. In the Ubl4A K48R cell line, depletion of USP13 still causes accumulation of ERAD substrate TCRα (data not shown). However, as we mentioned in our original manuscript, this cell line still contains endogenous Ubl4A, which is preferentially ubiquitinated when USP13 is depleted. To obtain a clean result, we need to knock down endogenous Ubl4A while maintaining the expression of ectopically expressed Ubl4A at a level comparable to that of the endogenous protein. This is technically challenging and will take a long time to complete (although we have started to play around with the CRISPR technology, we have not been able to establish a procedure to knock in a mutant gene). Additionally, it is also worth noting that although we describe ubiquitination of Ubl4A as a spurious event that damages the Bag6 function, we haven’t formally ruled out the possibility that transient ubiquitination of Ubl4A by gp78 may also contribute a positive function to ERAD, a possibility now mentioned in the Discussion. If this is true, one would not expect to rescue the ERAD defect in USP13 knockdown cells with the Ubl4A K48R mutant. Answers to these questions are important, but we feel that resolving these issues by new experiments will take an unreasonable amount of time and the required additional experiments are beyond the scope of the current study. We hope that the reviewers will agree with us on this issue.

*4) The direct link between gp78 and Usp13 is fairly weak and to a large extent depends on the analysis of Usp5/Usp13 chimera. In this respect, the statement that the second UBA-domain accounts for the differential activity between Usp5 and Usp13 is not really supported by evidence (this would need a more specific swap of UBA-domains, rather than large protein segments). Adding these data would clearly strengthen the link between gp78 and Usp13. Moreover, the effect of gp78-depletion on the Usp13/p97 interaction is very weak and only one shRNA against gp78 is shown; this experiment needs to be repeated with multiple shRNAs (or better, rescue constructs), allowing the researchers to obtain a potential correlation between strength of knockdown and phenotype*.

We now made an additional construct that contains mostly USP13 with only the segment downstream of the first UBA domain (totally 140aa) replaced by the corresponding sequence in USP5 (Figure 1). This construct, dubbed USP13/5-6 does not bind gp78 as efficiently as wild type USP13 (Figure 2). Collectively, the results clearly suggest that the second UBA domain is both necessary and sufficient for binding gp78.